



# In-flight calibration and monitoring of the TROPOMI-SWIR module

Tim A. van Kempen[1], Richard M. van Hees[1], Paul J.J. Tol[1], Ilse Aben[1,2], and Ruud W.M. Hoogeveen[1]

[1]SRON Netherlands Institute for Space Research, Sorbonnelaan 2, 3584 CA, Utrecht, the Netherlands
[2]Vrije Universiteit, Faculty of Science, De Boelelaan 1085, 1081 HV, Amsterdam, the Netherlands

**Correspondence:** Tim van Kempen (t.a.van.kempen@sron.nl)

**Abstract.** During its first year in operation the SWIR module of the TROPOMI instrument was calibrated in-flight and its performance was monitored. In this paper we present the results of the in-flight calibration and the ongoing instrument monitoring. This includes the determination of the background signals, noise performance, ISRF stability and stray-light stability. From these results, the number of incurred dead and bad pixels due to cosmic-ray impacts is determined. The light-path transmission is checked by monitoring internal lamp and diffuser stabilities. Due to its high sensitivity for Earth radiation on the eclips side, the calibration strategy for the background (i.e. dark flux and offset) monitoring was adjusted. Trends over the first full year of nominal operations reveal a very stable SWIR module with little to no degradation. The number of newly incurred dead and bad pixels is negligible since the start of operations. Assuming linear degradation of various component, the SWIR module is expected to keep performing within expected parameters for the full operational lifetime.

## 1 Introduction

The Sentinel-5 Precursor mission (Veefkind et al., 2012, better known as S-5P), is the first mission within the scope of the European Union Copernicus program[1] dedicated to mapping and monitoring of the chemical composition of the Earth's atmosphere. S-5P is a precursor mission for the atmospheric composition Sentinel-5 missions, which produces the same global coverage as S-5P. The Sentinel-5 mission are scheduled to launch in 2022 and beyond. The Tropospheric Monitoring Instrument (TROPOMI) is the sole instrument onboard S-5P. It consists of two modules: a UVN module (Veefkind et al., 2012) and a SWIR module[2] (Hoogeveen et al., 2013). The wavelength ranges include the spectral signatures of key trace gases that strongly influence climate and air quality. The SWIR module is aimed at measuring column densities of carbonmonoxide (CO) and methane ($CH_4$). TROPOMI will produce daily global coverage column density maps of these gases using a swath of approximately 2600 km across track. Images are taken each 1.08 seconds yeilding spatial pixels of approximately 7 by 7 km$^2$ at nadir. The SWIR spectral range (2305-2385 nm) is sampled at 0.1 nm, while the spectral resolution is 0.22 nm.

---

[1]see http://www.copernicus.eu
[2]The SWIR spectrometer was developed by SSTL under an Airbus-Dutch Space contract, with contributions of SRON and Sofradir.





With a total envisioned lifetime of 7 years, the mission will provide a unique insight into the chemical composition of our atmosphere. TROPOMI will be an essential tool to investigate both natural and anthropogenic induced chemical variations at timescales from days to years.

S-5P was launched on October 13th 2017, from Plesetsk, Russia, into an ascending sun-synchronous orbit with an equator crossing time at 13:30 Mean Local Solar time at an altitude of approximately 824 km. After launch, the first month was dedicated to the out-gassing the instrument; the S-5P cooler door was subsequently opened on November 7th 2017. In the following week, the SWIR detector and spectrometer cooled down to their operational temperatures of 140 K and 202 K respectively.

Between the launch and April 30th 2018, the commissioning phase, also referred to as the E1 phase, was carried out with the aim to complete the calibration of the instrument, check the data processing chain and prepare for the nominal operations phase, referred to as the E2 phase. Nominal operations started at orbit number 2818. During nominal operations, the instrument calibration needs to be monitored. This is done using measurements in the eclipse side of each orbit. TROPOMI covers the entire planet each day in 14.5 orbits. For the SWIR module, monitoring is performed for the background signal, the instrumental

noise, the quantification of the pixel quality and validation/monitoring of the instrumental spectral response function (ISRF) and stray-light correction. Correction are based on so-called Calibration Key Data (CKD). The correction algorithm, on-ground calibration and CKD (Calibration Key Data) derivations of the ISRF is in van Hees et al. (2018, ISRF) and of the stray-light correction in Tol et al. (2018, Straylight). CKDs for offset, dark-flux, noise and pixel quality were also derived on-ground. Signals of the sun as seen over the two diffusers and signals of the internal lamps are monitored to quantify any transmission

changes of various components within the SWIR module.

In this paper we will report on the results of the commissioning phase, the monitoring during the first full year of nominal operations and provide an outlook of the durability and future performance of the SWIR module. The outline of the paper is as follows. Section 2 details the calibration plan. Section 3 presents the results of the commissioning phase. Section 4 describes

the monitoring results and trends of the first year of TROPOMI. Finally, the conclusions are given in Section 5.

## 2   In-flight Calibration and Monitoring Plan

### 2.1   Calibration Plan

The calibration of SWIR is done primarily using data obtained during on-ground calibration campaign (Kleipool et al., 2018). It is a key part of the calibration plan to monitor the quality of these on-ground calibrations over the lifetime of TROPOMI and

update procedures and/or the CKDs if necessary.

There are several types of measurements available for in-flight calibration:

–   radiance (i.e. backscattered radiation from the Earth, both for the day and night-side)





- irradiance (i.e. radiation from the Sun)

- measurements with a closed folding mirror mechanism (FMM), looking into the on-board calibration unit (CU).

When the FMM is closed, the SWIR module can be illuminated by several on-board calibration sources installed specifically for in-flight monitoring of calibration parameters. In addition to the background measurements (i.e. all sources turned off), the following on-board illumination sources are relevant for the SWIR module:

- DLED - a dedicated LED emitting with a known smooth spectral profile at the SWIR wavelengths.

- WLS - White Light Source

- SLS - Spectral Line Source: five dedicated diode lasers in the SWIR spectral band

The DLED is placed in front of the detector behind the immersed grating, while the SLS and WLS are located in the Calibration Unit and thus follow almost the complete optical path. This is an important difference to distinguish effects of the full optical path, or of the detector only.

The five on-board tune-able distributed feedback lasers, or SLS, are unique to the SWIR module. These lasers are able to scan small parts of the wavelength range by changing the laser temperature using a thermo-electric cooler integrated into the laser housing. The range is about 70 detector pixels (∼7 nm), although due to operational restrictions typically a much smaller scan is done of about 6 detector pixels (∼0.6 nm). The central wavelength of each laser has been selected to be able to sample different parts of the SWIR wavelength range. The signal of the SLS passes over one of two diffusers of the CU. Each diffuser can be employed in oscillation mode to suppress speckles observed in the laser signal. Due a limited operational lifetime and excess heat produced by the oscillating diffuser, it was decided to not oscillate the diffuser during nominal operations (van Hees et al., 2018).

Table 1 lists the parameters for which calibration parameters, the CKD, or monitoring data are derived. Measurements are taken in-flight to monitor whether the CKD can still be applied correctly during data processing. Other quantities are monitored, but do not have a direct relation to CKD parameters. Monitoring of these quantities is essential for the health monitoring of the SWIR module.

## 2.2 Processing Chain

Science signals of TROPOMI SWIR are taken from a detector array consisting of 1000 pixels in the spectral dimension and 250 pixels in the spatial dimension. Each pixel is read out individually, but the exposure time is identical for all pixels in the detector. Exposure times during nominal operations range from 82 ms to typically 1080 ms. Shorter exposure times are used to avoid detector saturation in case of high input light levels. For reference to be used in the remainder of the paper: a pixel signal can be between 0 and 500,000 electrons, leading to electrical signals between 0.5 and 3.5 Volts, digitized typically with 12,000 Binary Units (BU). A raw TROPOMI-SWIR signal consists of three components: an offset, which is independent of exposure time; a dark signal, which is dependent on exposure time; and an outside signal. Outside signal can either be the Earth radiance,





| Quantity | CKD Type | Measurements |
|---|---|---|
| Dark Flux | Static | Dark |
| Offset | Static | Dark |
| Noise | Static[1] | Dark |
| Quality[2] | Static | - |
| Lamp Stability | Monitor | DLED/WLS/SLS |
| PRNU[3] | Static | DLED/WLS |
| Diffuser Stability | Monitor | Irradiance |
| Transmission | Monitor | DLED/WLS/Irrad. |
| ISRF | Monitor [4] | SLS |
| Stray-light | Static | SLS |

**Table 1.** Calibration and monitoring data obtained in-flight

[1] The Noise CKD is static. However, the in-flight noise of the detector is measured dynamically as input for the quality. Readnoise was derived on-ground.

[2] The quality map does not use direct measurements, but uses the dynamically measured dark flux and in-flight noise.

[3] The PRNU stability is included in the comparison of the different lights

[4] The ISRF is not used in the L1b data processing, but used in the SWIR retrievals such as CO or $CH_4$.

Solar irradiance or signal from the on-board lights. To accurately derive the useful signal (i.e., the outside signal) the offset and dark flux signals must be determined to high precision and in turn subtracted from the raw signal. To calibrate the useful signal, the outside signal has to be corrected for with several factors that influence the signal, such as the transmission (i.e., amount of light lost), pixel response non-uniformity (PRNU) and influence of stray-light. Stray-light is defined as any outside

5     signal that does not follow the intended path onto the detector and is thus not part of the useful signal. Stray-light correction for the SWIR module is extensively discussed in Tol et al. (2018). In-flight stray-light monitoring is discussed in Section 3.5. Hoogeveen et al. (2013) mentions a few other effects as observed in the SWIR detector. A small pixel-memory correction is only applied when the exposure time is 1080 ms. With faster detector readout, data are co-added, making the memory error smaller, and more difficult to correct for. Given the range of typical exposure times, non linearity of the detector was judged

10     to be too small to justify a complex correction algorithm The wavelength calibration is not specifically monitored, but follows from trace gas retrieval algorithms where small wavelength shifts are fitted within the procedure. Figure 1 shows a flow chart of this processing.





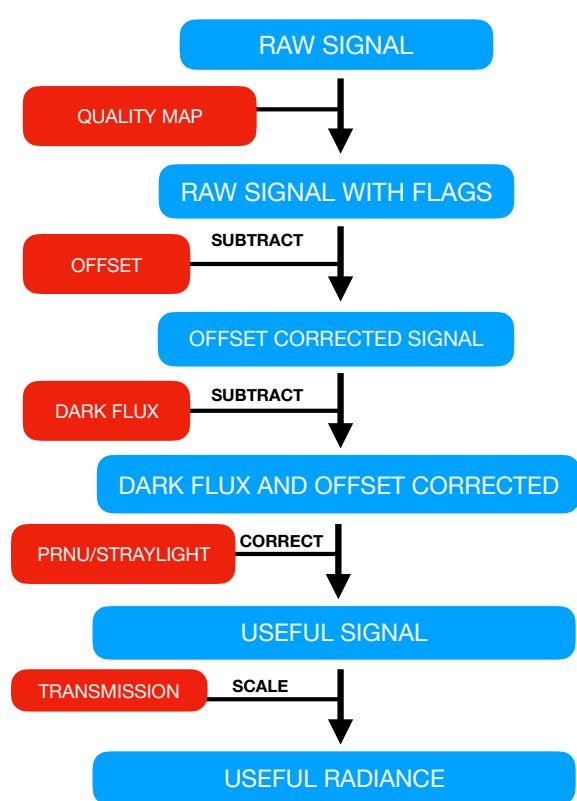

**Figure 1.** Flow chart of the processing chain of a SWIR signal.

## 3 In-flight Calibration during the commissioning phase

### 3.1 Dark Flux and Offset

#### 3.1.1 Method

The value of the offset en dark-flux corrections are determined from measurements at the eclipse side of the orbit, see Table 1.
Measurements are carried out with identical instrument settings (exposure time and co-adding factor) as the radiance measurements on the solar illuminated side of the orbit. The exposure times range from 178 ms over the equator to 538 ms over the poles. Before launch, it was assumed the eclipse side of the Earth is dark and the raw signal is composed only of the offset and dark flux. A linear fit using measurements at a range of exposure times will yield the offset (signal at exposure time zero) and dark flux (slope of the fit). In total, derivations are done every 15 orbits, using all background measurements within those 15 orbits.





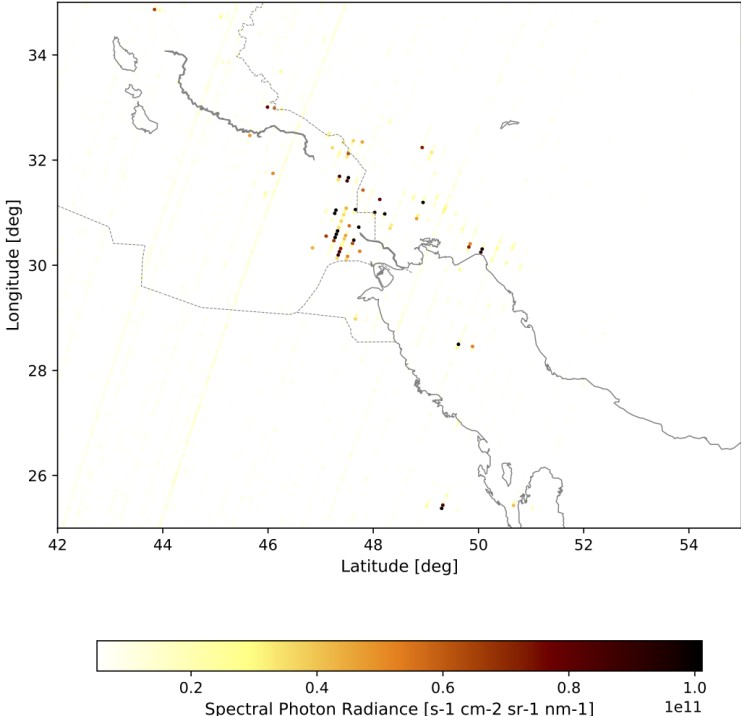

**Figure 2.** Continuum radiance at 2314 nm at the eclipse side of the Earth around the Iraqi city of Basra Any stripes are limitations of calibration and thermal stability at the time. Localized enhanced signals are clear indications of emission sources on the Earth.

### 3.1.2   Background with FMM open

Figure 2 shows the radiance of the SWIR continuum at 2314 nm at the eclipse side of the orbit around two regions: the northern part of the Persian Gulf and north-western Australia. Data was taken from orbits 430 and 433, measured during the first-light campaign during November 2019. All exposure times were 216 ms.

5    In both scenes of Figure 2 and 3 small regions and point sources are clearly visible with signals more than an order higher than the background. Given the location, the sources are most likely the burning of natural gas at oil field installations (Basra) or natural wildfires (Australian outback). Inspection of other data yields many other emission sources over land including other bush fires and volcanic activity.

At larger spatial scales, thermal radiation of the Earth at night is detected by the SWIR module both over land and over
10    oceans. Thermal radiation of the oceans appears brighter, presumably due to inherently longer cooling times of water. However, even at high latitudes, radiances are clearly nonzero at the eclipse side of the orbit.

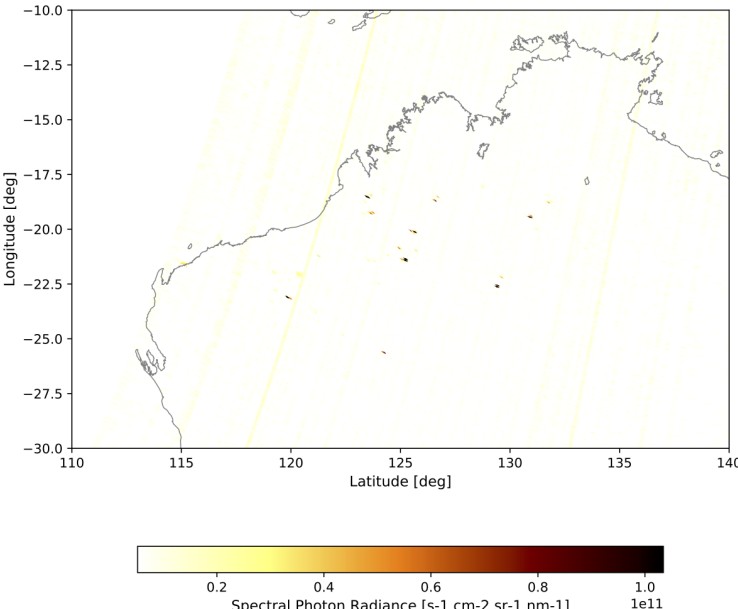

**Figure 3.** Continuum radiance at 2314 nm at the eclipse side of the Earth for the north-western Australian outback. Any stripes are limitations of calibration and thermal stability at the time. Localized enhanced signals are clear indications of emission sources on the Earth.

Figure 4 presents the dark flux at detector level using measurements with the FMM open. The top plot shows the results, while the bottom row shows the difference with the dark flux derived from the on-ground calibration. Statistics of the results over the entire detector, i.e. the biweight median and spread[3], are given in Table 2. The comparison reveals the following:

–  The overall structure of the dark flux on the detector is reproduced, see Hoogeveen et al. (2013).

5   –  The median over the detector is somewhat lower (61 e/s).

–  The difference in spreads is significant due to the amount of data used in obtaining the results.

–  Specific spectral features can be seen in the comparison to the on-ground calibration Fig. 4 in the form of blue bands. The wavelengths correspond to deep absorption bands of water and methane. Presumably, absorption of the Earth's thermal radiation by water and methane occurs causing this difference.

10   Analysis of a range of measurements over 3 months also revealed differences with the on-ground calibration results to vary in time by ∼20 e/s. No trend was seen, but local changes in e.g. Earth's temperature field or weather can influence the results. These are partially mitigated by taking a bi-weight median over all available data, but this method cannot completely remove these effects.

---

[3]Throughout this paper, the biweight median and biweight spread are used. For simplicity the terms median and spreads are used throughout. Biweight median is a statistical parameter described in Beers et al. (1990).





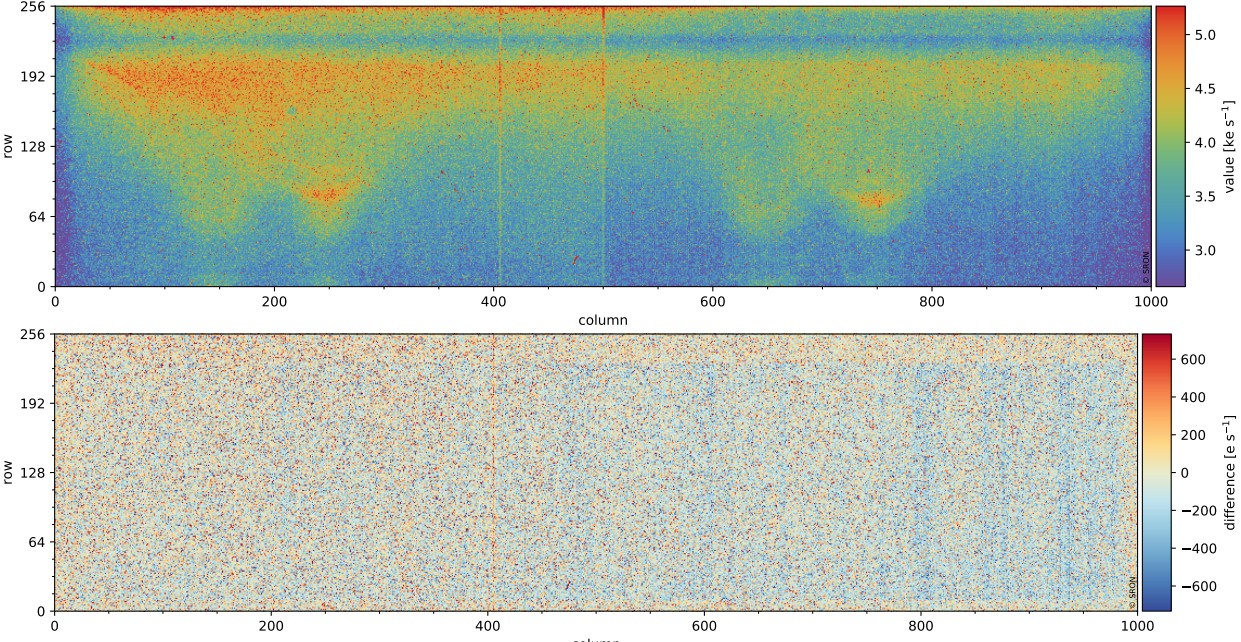

**Figure 4.** Typical dark flux obtained with FMM open during the commissioning phase using data from orbits 990 to 1004 is seen in the top plot. Data is plotted over the detector with the horizontal axis equivalent to the spectral direction and the vertical axis the spatial swath. The bottom plot shows the result with the on-ground results subtracted.

The amount of dark flux detected differs from the measured value reported in Hoogeveen et al. (2013). They present a median in the central area of 0.7 fA, equivalent to 4400 electrons per second. This is likely attributed to thedifferent thermal conditions of the setup, as a significant part of the dark flux is caused by the thermal emission of the spectrometer, which was absent in Hoogeveen et al. (2013).

5    The non-uniformity of the Earth's thermal radiation also introduces another significant bias. As most calibration measurements are taken near the warmer equator, the measurements are not representative for the complete orbit including the polar regions.

| Origin | Orbit | Median | Spread | On-ground Diff. |
|---|---|---|---|---|
| | | [e/s] | [e/s] | |
| FMM Open | 1004 | 3736 | 14.4 | 61 |
| FMM Closed | 2721 | 3772 | 20.3 | -25 |
| Nominal Operations | 7778 | 3764 | 16.5 | -33 |

**Table 2.** Median and spread of the dark flux and comparison with the median of on-ground calibration. The on-ground results are based on many more measurements. As such the uncertainties calculated from the spread are not comparable.





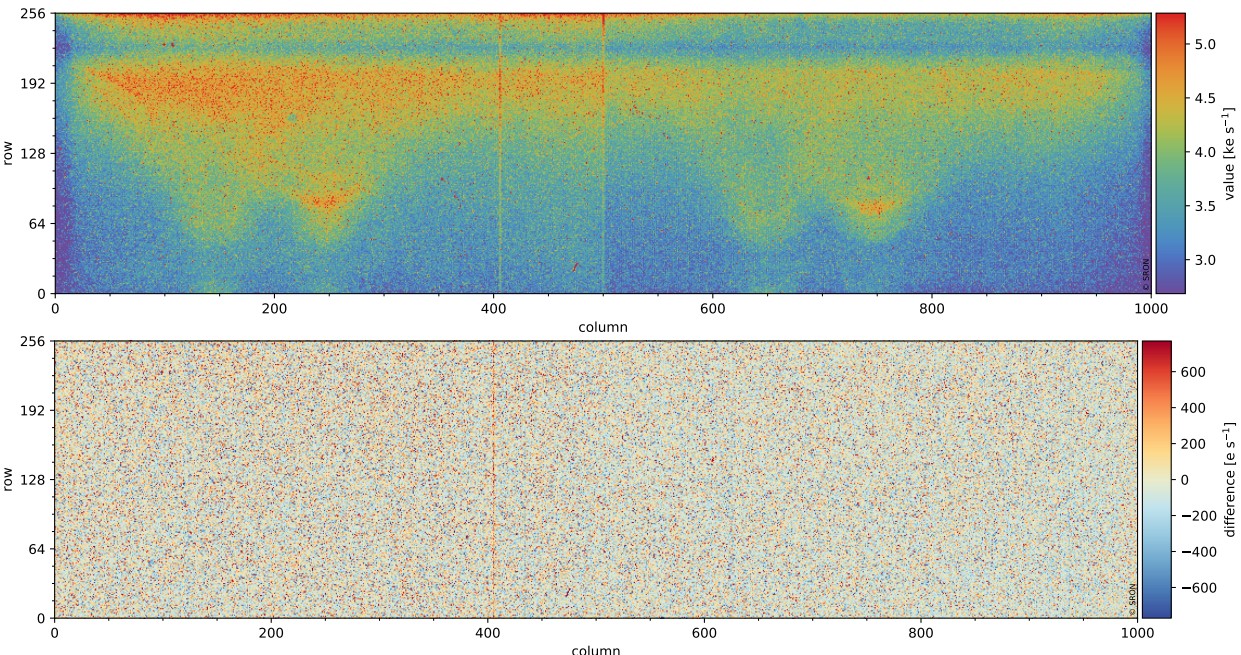

**Figure 5.** Typical dark flux obtained with FMM closed during the commissioning phase using data from orbits 2818 to 2833 is seen in the top plot. Data is plotted over the detector with the horizontal axis equivalent to the spectral direction and the vertical axis the spatial swath. A comparisons to the dark flux derived during the on-ground calibration is shown in the bottom plot.

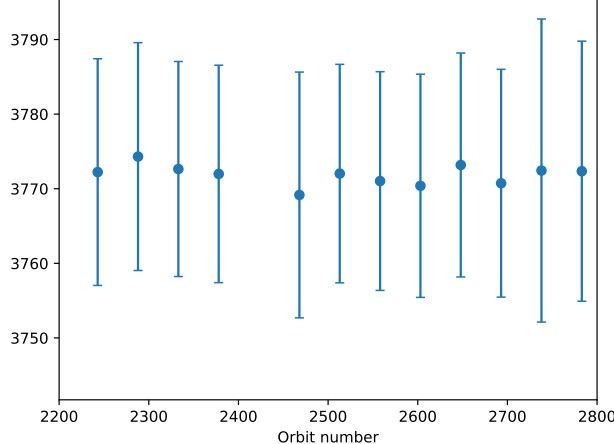

**Figure 6.** Median dark flux and its uncertainty obtained with FMM closed during the commissioning phase between orbits 2200 to 2800 (end of the commissioning phase) as a function of time. The shown timelength is about six weeks. Note that due to the structure seen in the dark flux, the spread (not shown) is much larger.





### 3.1.3 Dark Flux with FMM closed

Given the issues described in section 3.1.2, background measurements were also performed with the FMM closed.

Figure 5 shows the derived dark flux with the FMM closed, and its comparison with the on-ground result. The detector median is given in Table 2. With the FMM closed, dark flux does not differ significantly with the on-ground results.

Averaged over the entire detector, the dark flux is lower by ∼25 e/s than the on-ground measurements. This is likely due to different thermal conditions. Another difference with the on-ground results is found in the spread (i.e. the uncertainty of the fit). This is caused due to the number of input points for each fit. Both the number of different exposure times as well as the amount of the measurements available for each exposure time was higher during the on-ground calibration. However, the detected systematic differences between on-ground and in-flight with the FMM open, such as the absorption bands or latitude

dependent signal are clearly absent when the FMM is closed.

The dark flux with the FMM closed was also tracked in time over the last two months of the commissioning phase. Derivations were carried out at intervals of 15 orbits with the requirement that at least 40% of all orbits contained background measurements. Figure 6 reveals that the dark flux with the FMM closed is very stable with variations of 2-3 electrons per second from derivation to derivation.

### 15 3.1.4 Orbital Dark

During nominal operations, measurements are typically taken at northern latitudes of the eclipse side of the orbit. As such, any variation within a single orbit cannot be monitored or calibrated. This may lead to a systematic error if there are thermal variations within a single orbit. Accurate calibration of the dark flux thus includes a calibration of thermal variations as a function of the orbital phase, using background measurements over a several orbits with the FMM closed. The observed signal

with the FMM closed as a function of orbital phase was inspected at exposure times of 100, 500 and 1000 milliseconds. The data show no dependency of the dark flux over the orbit. Therefore, no orbital variation of the dark-flux correction is applied in the data processor. Two increases in the signal were detected, both during overpasses of the South Atlantic Anomaly (SAA) region. Within the SAA, the van Allen radiation belt dips much closer to the surface of the planet, significantly increasing the amount of cosmic radiation hits on the detector and thus to a small increase in average background signal. No correction is

applied for the SAA in the data processor. Instead, all measurements in the SAA are flagged to be less reliable.

### 3.1.5 Conclusions on FMM setting

In conclusion, background measurements with the FMM closed produce more accurate and more stable dark fluxes than measurements with the FMM open. Surface features, such as fires and the land/sea difference, are removed from background measurements if the FMM is closed. In addition, the accidental introduction of spectral features due to methane and water

absorption in the thermal radiation is also removed. If the data taken in the SAA are excluded from the analysis, no orbital dependency of the dark signal is necessary. Dark flux is also shown to have similar values as measured during the on-ground calibration as well as the values reported in the detector characterization (Hoogeveen et al., 2013).





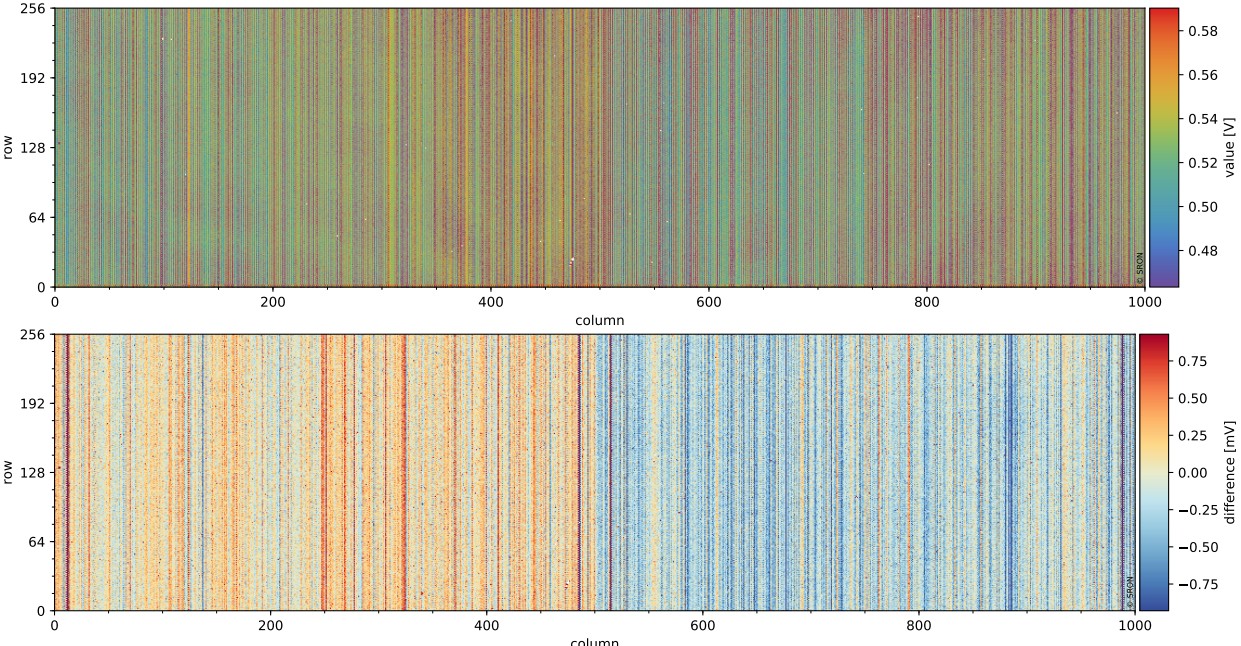

**Figure 7.** Offset obtained with FMM closed during the commissioning phase using data from orbits 2707 to 2721 is seen in the top plot. Data is plotted over the detector with the horizontal axis equivalent to the spectral direction and the vertical axis the spatial swath. A comparison to the offset derived during the on-ground calibration is shown at the bottom plot.

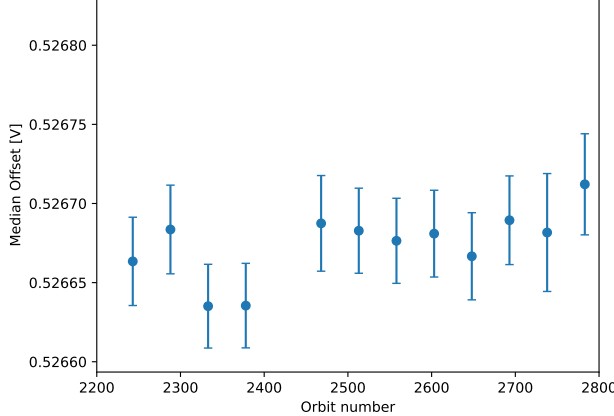

**Figure 8.** Median offset and its uncertainty obtained with FMM closed during the commissioning phase between orbits 2200 to 2800 as a function of time.



### 3.1.6 Offset

Since the offset is derived using the same set of measurements as the dark flux, the offset of the SWIR detector shows similar dependencies as the dark flux. All conclusions for the dark flux also apply to the offset. Figure 7 and 8 show the offset with the FMM closed and its dependency on time as a reference. Note that there appears a small systematic difference between the ADCs, which is not well understood, but currently attributed to different thermal conditions. This difference is well below acceptable levels.

### 3.2 In-flight Noise

The noise on all signals read is composed of three components: (i) shot noises of the external signal, thermal background and dark flux, (ii) Johnson noise and (iii) readnoise. These combine to form the in-flight noise. Readnoise is independent of exposure time, while the other noise components depend on the exposure time. Readnoise was calibrated during the on-ground calibration campaign by measuring the noise versus exposure time and extrapolating back to zero exposure time. The other noise components are grouped as in-flight noise. It is necessary to measure the in-flight noise of each pixel without any external signal or its shot noise as input for the detector pixel quality monitoring. Detector pixels with too high noise levels (either readnoise or in-flight noise) should not be used for retrieval of CO or $CH_4$ and are flagged in the processor.

Early in the commissioning phase, in-flight noise calibration measurements were executed with the FMM open, similar to the dark flux and offset. However, similar effects were seen for the noise as discussed in section 3.1, with signals from point sources and non-uniform Earthshine influencing the noise derivations. Therefor, calibration measurements to determine noise levels should also be executed with the FMM closed.

Figure 9 shows the in-flight noise with the FMM closed, taken 6 months after launch. Noise can be derived either by taking the standard deviation over all frames within a measurement, with the median subtracted, or the spread of all frames. For a symmetric gaussian distribution of the data points, both methods yield an identical result. But for a skewed distribution with outliers, the standard deviation method tends to result in a higher noise than the bi-weight median. In Figure 9, both are plotted. For the SWIR module, outliers are produced by cosmic ray impacts that manifest themselves as dots and small tracks in Figure 9.

Figure 10 shows the comparison between the readnoise CKD as measured on-ground and the one measured in-flight. Figures are shown using derivations with a standard deviation and a bi-weight spread, highlighting the impact of cosmic rays.

### 3.3 Detector Pixel Quality

The quality map of the SWIR detector details how many detector pixels are of sufficient quality to be included in retrieval algorithms. In the definition of ′sufficient quality′, a pixel should

– have a proper response to light, as determined by a linear response to light as a function of exposure time.

– not show excessive noise

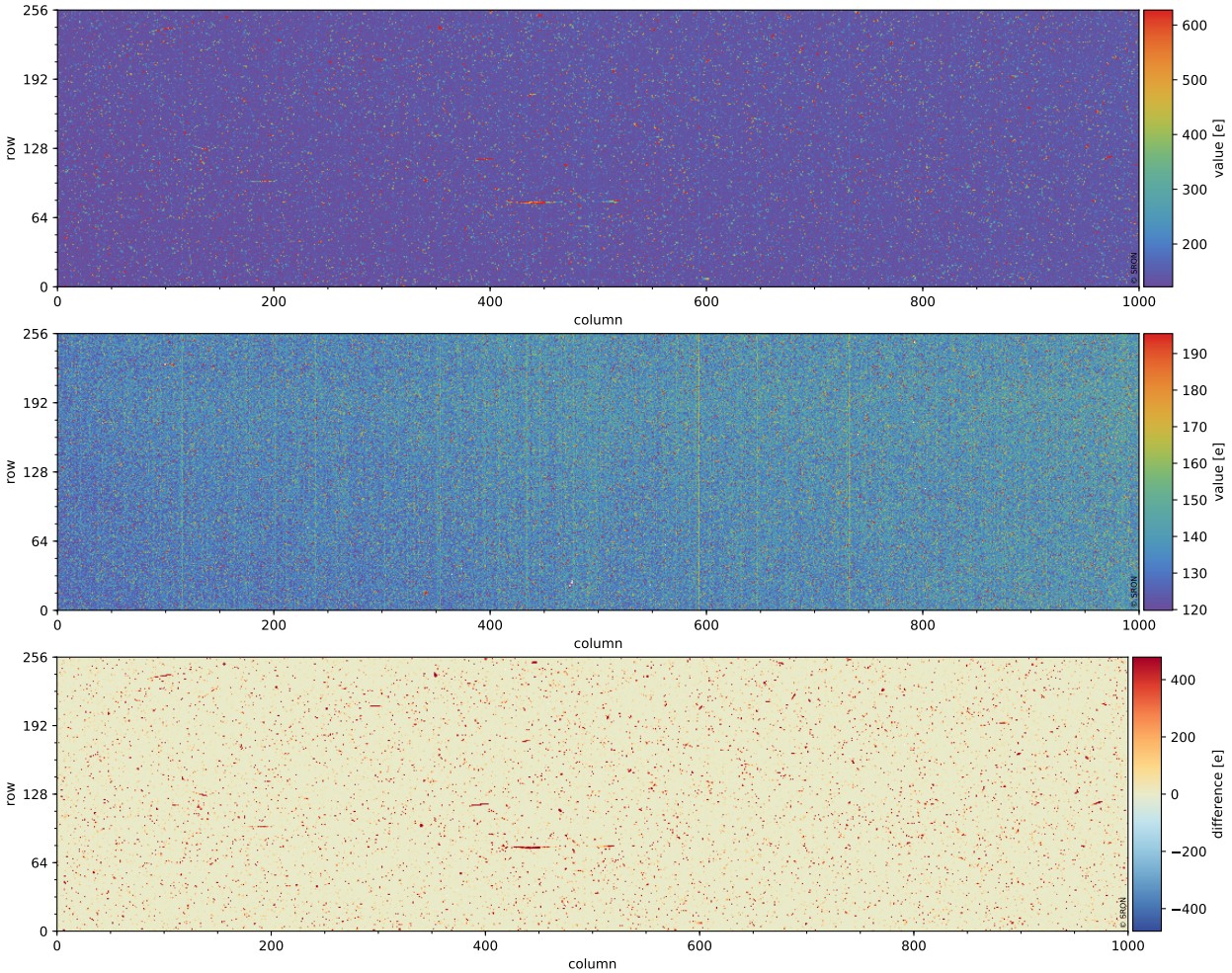

**Figure 9.** Noise of the SWIR detector measured during orbit 2707 to 2722 for an exposure time of 538.3 ms with the FMM closed, showing the noise calculated as a mean (top), median (middle) and the difference between the two (bottom).

– not produce excessive dark flux

Using a weighted function (for which the weighting was determined using on-ground calibration measurements), each pixel is graded with a number between 0 (completely dead or unusable pixel) and 1 (perfectly working) using measurements of the noise and dark flux. A detector pixel is considered to be bad if this value is lower than 0.8. A ′dead′ category is tracked by considering detector pixel with values below 0.1. This includes pixels with no response (i.e. a value of 0.0). If required, manual flagging is also possible within the processor (i.e. setting this quality to 0.0). Some pixels were known to be not functional even before launch. This includes pixels outside of the effective area which are not illuminated. Other may become unusable over time (e.g., no signal, too noisy) due to cosmic ray impacts or other hardware degradation.



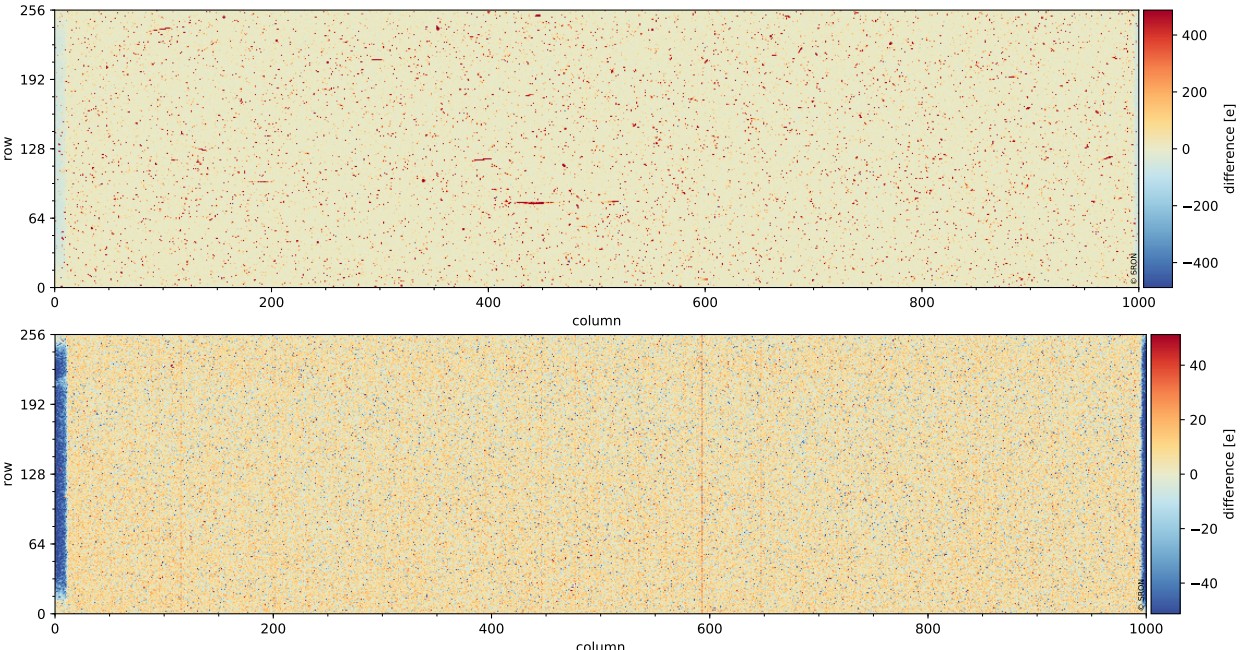

**Figure 10.** Comparison between the noise as measured in-flight during orbit 2707 to 2722 and on-ground. The top plot shows the difference between derivations using a root mean square while the bottom shows that using a bi-weight spread. Note that the similarity of the difference between on-ground and in-flight and the difference between the two methods as shown in the bottom part of Fig. 9.

.

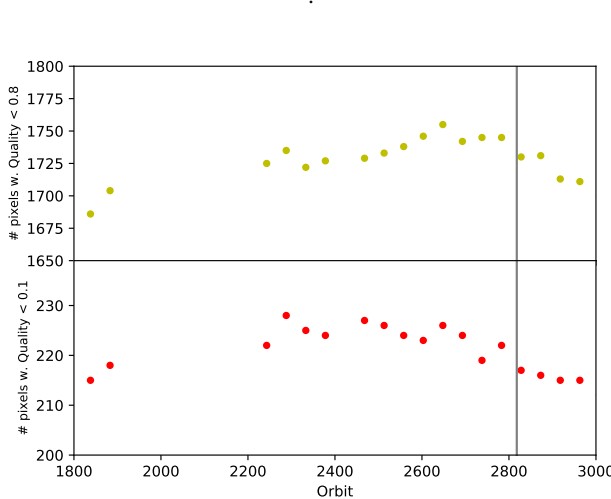

**Figure 11.** Number of dead and bad pixels. Pixel quality is expressed as a number between 0 and 1.

Figure 11 shows the number of flagged pixels at the end of the commissioning period. Only data from orbit 1800 and later was analysed as data was taken in a consistent procedure with the FMM closed. An open FMM, heavily influenced the dark





and noise, and thus the quality. This quality map is derived using a bi-weight median noise, given the limitations discussed above. Table 3 lists the number of pixels identified both on-ground, during the the commissioning phase and at the start of nominal operations. Note that the quality is derived using all available data once per 15 orbits.

| Origin | bad quality | dead quality |
| --- | --- | --- |
| on-ground | 2283 | 258 |
| orbit 1838 | 1686 | 215 |
| orbit 2828 | 1730 | 217 |

**Table 3.** Number of detector pixels labelled as 'bad' (quality $< 0.8$) or 'dead' (quality $< 0.1$). Note that this covers the total of 250,000 detector pixels. The area used for retrieval equals $\sim$210,000

## 3.4 Transmission

The stability of the transmission of the optical components is checked by comparing the signal of various on-board calibration sources and the solar irradiance measured with the on-board diffusers. Although monitoring of the transmission of the full optical train for radiance measurements is the main goal, it can only be approximated with the methods applied. Changes seen in the signal of the calibration sources and/or solar irradiance signals can originate from degradation of the sources and/or diffusers. Both the calibration sources and diffuser are expected to degrade over the operational lifetime. After cross-calibration,

any changes in the transmission should be carefully monitored and investigated. In this section we will compare the output of the on-board calibration sources and compare it to the results obtained on-ground.

### 3.4.1 DLED

The DLED is intended to monitor the stability of the detector. In-flight, monitoring of the detector signal caused by the DLED illumination is done by comparing the DLED response to a reference measurements taken late in the commissioning period.

The reference measurement has in turn been calibrated to the on-ground reference

Figure 12 shows the measurement of orbit 907 (Dec 2017) and 2707 (Apr 2018) as compared to the reference measurement, which was set to the measurement in orbit 2515. The DLED responses as seen in Figure 12 already show that the DLED has degraded between orbits 907 and 2707, relative to the reference orbit of 2515. However, typical degradation is seen at a level of 0.1%. Features in the measurement of orbit 907 appeared after launch, and vanished, which is not well understood. It is

hypothesized this is influenced by the thermal stability of the grating. The degradation is further discussed in section 4.5.





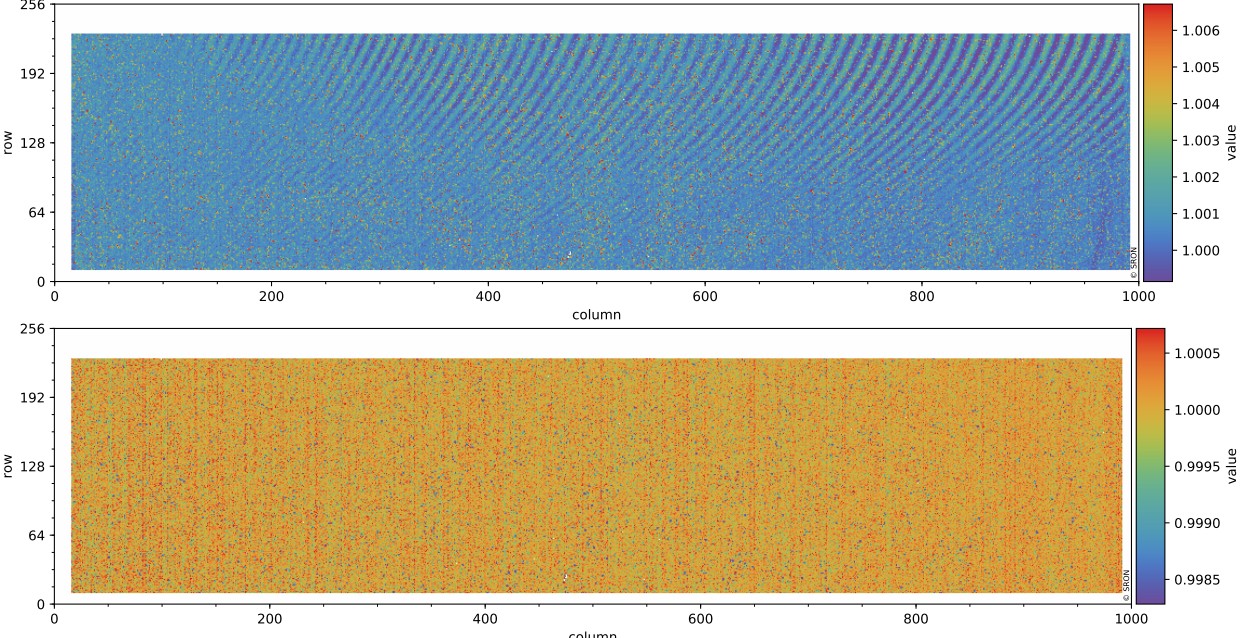

**Figure 12.** DLED response of orbits 907 and 2707, relative to the in-flight reference of orbit 2515.

### 3.4.2 WLS

A Tungsten halogen lamp is mounted inside the Calibration Unit and act as a white light source (WLS). Its output follows the complete light path within the module with the exception of the primary mirror. The main driver is that the WLS settings, including the power, are optimized to quantify its response at UV and UVIS wavelengths.

5  The WLS settings have been optimized to yield sufficient signal in the UV and UVIS wavelengths. This results in a relatively strong output in the SWIR wavelength band. To avoid saturation, a short exposure time (5 ms) for SWIR has to be applied. A reference for the WLS was derived at the end of E1. Given the much less stringent stability limits of the WLS system, no differences were found in the resulting SWIR signals that were not seen in the SWIR signals due to the DLED. This indicates that the optics of the SWIR module is stable over time.

### 10 3.5 Stray-light

The methodology to determine the stray-light calibration key data, including the on-ground measurements used, are described in detail in Tol et al. (2018). In-flight there is no capability to directly quantify the amount of stray-light within the SWIR module as a response to a point source illuminating any location of the SWIR detector. However, there is a possiblity to monitor the stability of the stray-light CKD over time by comparing the signal response of one of the on-board diode lasers

15 with the equivalent on-ground measurement. Before launch, SLS-1 was selected for regular calibration measurements as its wavelength is located near the center of the SWIR band. The effectiveness of the stray-light CKD is checked by comparing the





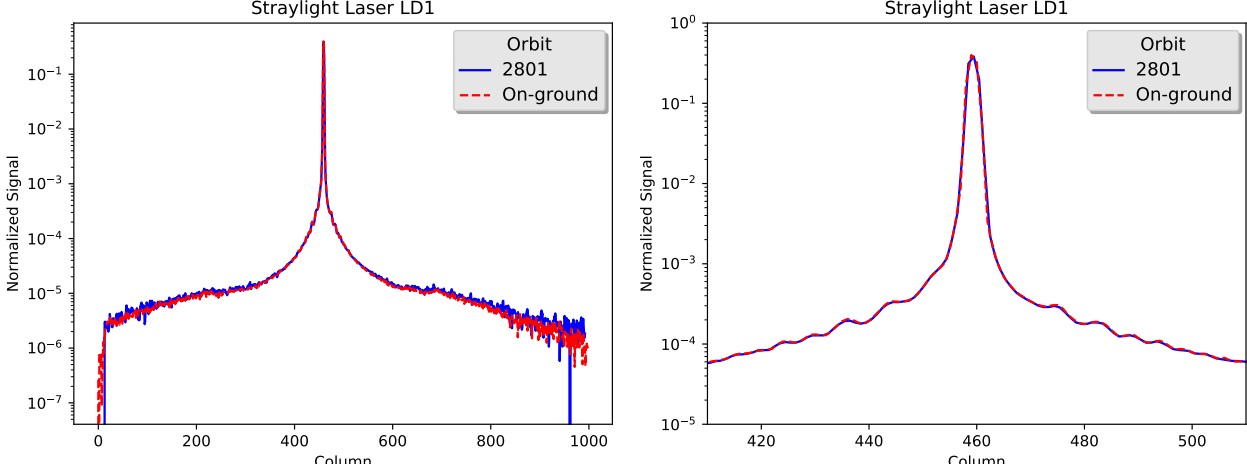

**Figure 13.** Comparison of in-flight measurements using SLS-1 during a single orbit with an identical measurement obtained during on-ground calibration. *Left plot*: full image. *Right plot*: zoom near the wavelength of SLS-1.

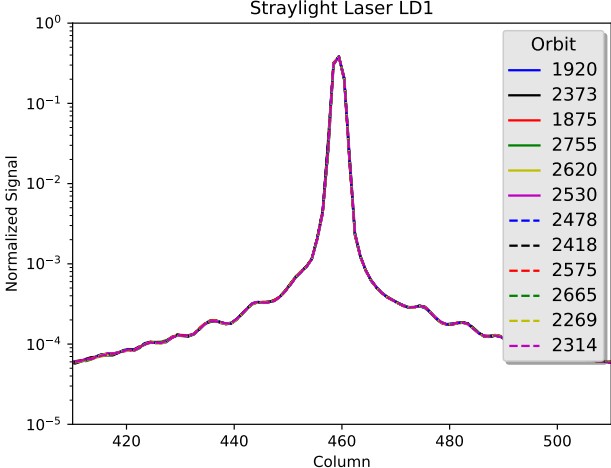

**Figure 14.** Results of all stray-light measurements with SLS-1 during the the commissioning phase.

known signal response of SLS-1. Monitoring is done by merging a short (98 ms) and long (1998 ms) exposure. Frame merging is discussed in Tol et al. (2018), section 3. This method produces a frame with an unsaturated line centre while still retaining good signal-to-noise outer wings.

   Figure 13 shows the spectral axis with medians taken over the swath for the on-ground and in-flight measurements for SLS-
5 1. Both the full dynamic range and a zoom around the laser wavelength are shown. This reveals no changes before and after launch in the distribution of the stray-light near the laser peak.



| Orbit | Amount Stray-light [%] | Uncertainty [$10^{-2}$%] |
|---|---|---|
| 1875 | 2.90 | 2.3 |
| 1920 | 2.89 | 2.2 |
| 2269 | 2.91 | 2.2 |
| 2314 | 2.90 | 2.2 |
| 2373 | 2.93 | 2.3 |
| 2418 | 2.88 | 2.1 |
| 2478 | 2.89 | 2.3 |
| 2530 | 2.89 | 2.2 |
| 2575 | 2.91 | 2.2 |
| 2620 | 2.90 | 2.2 |
| 2665 | 2.89 | 2.2 |
| 2710 | 2.90 | 2.3 |
| 2755 | 2.93 | 2.3 |
| 2801 | 2.90 | 2.1 |

**Table 4.** Percentage of light detected on the SWIR detector outside of the central 15 pixels of a response to diode laser SLS-1 during the commissioning phase.

Figure 14 shows all measurement taken with SLS1 during the the commissioning phase, overplotted onto each other. This reveals that the amount and shape of stray-light has remained stable over the course of the first few months after launch. This is confirmed by the tracking of the amount of stray-light, seen in Table 4. The amount of stray-light is defined as all light seen outside the 15 spectral pixels centered on the laser peak. Note that this is not a direct quantification of the stray-light, but

suffices as a monitoring quantity for the amount of stray-light.

### 3.6   ISRF

The Instrument Spectral Response Function (ISRF) of each pixel is required as input data for the gas-retrieval algorithms. The complete method to derive the ISRF CKD is described in van Hees et al. (2018). The CKD was derived using on-ground calibration measurements using an external tunable laser with the capability to illuminate limited parts of the swath. van Hees

et al. (2018) proposed a method to monitor the ISRF using the on-board diode lasers. Each diode laser illuminates a different area on the SWIR detector (and thus probes different parts of the ISRF spectral parameter range). A local "monitoring" ISRF is derived from these measurements. As the diode laser illuminates the full spatial swath and the five lasers only sample very small ranges of the full spectral axis, the diode lasers cannot be used to derive ISRF CKD. Their use is to detect and monitor long-term changes in the ISRF, if any. During the commissioning phase, the results from van Hees et al. (2018) were verified

to determine any possible changes between on-ground calibration and E1 performance. At the same time, a checkout was

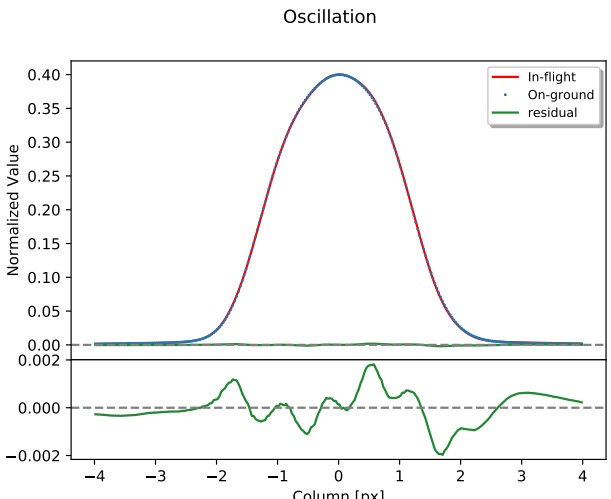

**Figure 15.** Comparison between the ISRF measurements taken during the on-ground calibration campaign and the commissioning phase. *Top plot*: Normalized pixel response in-flight (red) and on-ground (blue). In green, the residuals are plotted. *Bottom plot*: Zoom of the residuals.

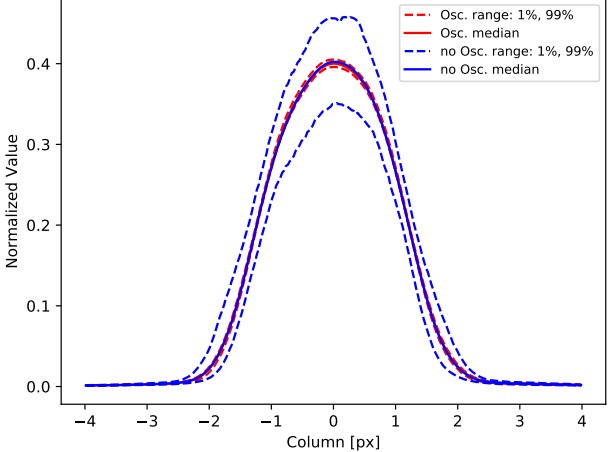

**Figure 16.** Comparison between the ISRF measurements taken with and without an oscillating diffuser. In red the measurements with an oscillating diffuser are plotted and in blue measurements without an oscillating diffuser.

performed of the diode laser settings for use during nominal operations (E2). In this paper results from diode laser SLS-1, which is the main reference during nominal operations, is presented, but conclusions apply to results obtained for the other four diode lasers.

Figure 15 shows the difference between measurements using diode laser SLS-1 carried out on-ground and in-flight. These

5    were done using identical settings. In this figure, the normalized pixel response is compared by taking a median over the



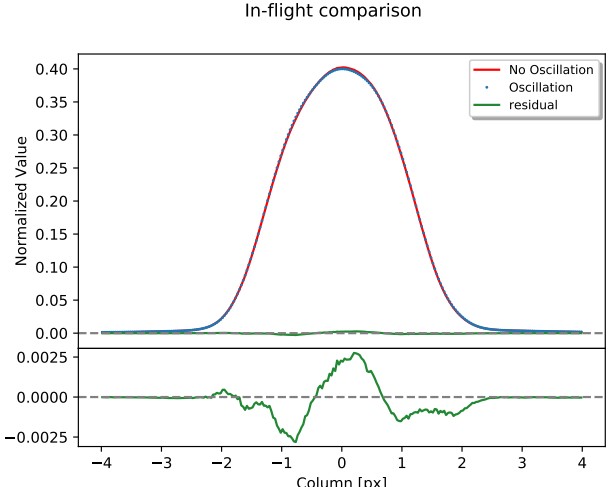

**Figure 17.** Comparison between the ISRF measurements taken with and without an oscillating diffuser. In-flight measurements during nominal operations are done without an oscillating diffuser. *Top plot*: Normalized pixel response in-flight (red) and on-ground (blue). In green, the residuals are plotted. *Bottom plot*: Zoom of the residuals.

illuminated swath (220 rows) and normalizing over the total energy. The difference observed between in-flight and on-ground is less than 0.2% indicating no change of the instrument between on-ground calibration and E1.

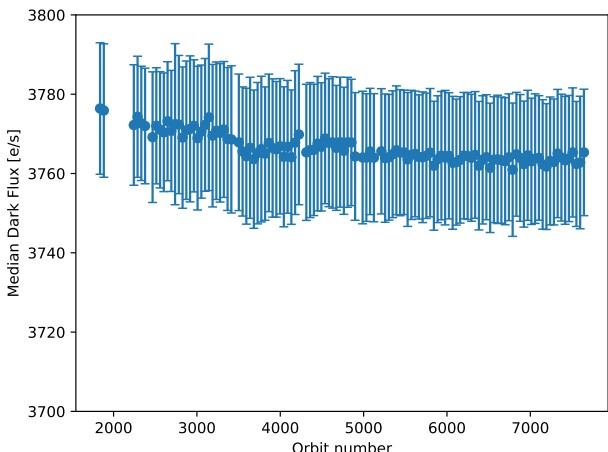

**Figure 18.** Median Dark Flux as a function of time.

During nominal operations, diode laser measurements are carried out using a fixed diffuser instead of an oscillating diffuser. The oscillation is needed to randomize the speckles of the monochromatic laser. However, as the diffuser motion is a life limiting item producing too much excess heat, it cannot be used during regular E2 monitoring. The resulting speckle pattern can be



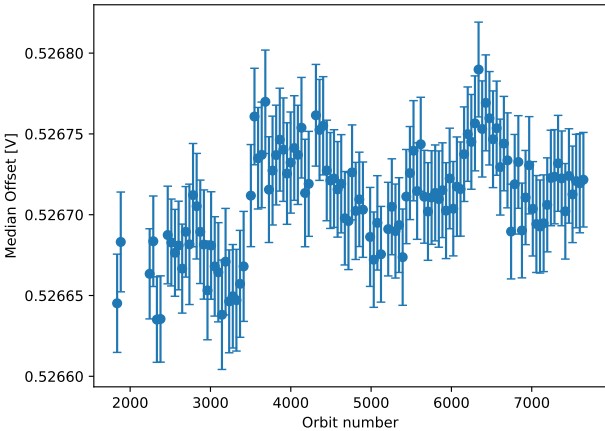

**Figure 19.** Median Offset as a function of time.

partially randomized by taking the median signal over all 220 rows illuminated. Figure 16 shows a comparison between in-flight measurements with identical settings except the (lack of) oscillation of the diffuser. In this plot the percentile range between 1 and 99 % is shown for all ISRF solutions in the spatial direction. It is clear that the range of solutions is much larger without an oscillating diffuser. However, the median ISRF solutions of both sets of measurements are very similar. Figure 17

shows this difference. From this figure it can be confirmed that the measured profile without an oscillating diffuser, although less accurate than the actual ISRF, is of sufficient quality to monitor the stability of the ISRF calibration in-flight. The figures in this section reaffirm that no changes larger than 0.2% are seen.

## 4   Monitoring Results during nominal operations

After the start of nominal operations on April 30th, 2018, the performance of the SWIR module has been closely monitored.

In this section, the trends are determined averaged over the full detector (i.e. either a mean or median of the properties over the detector or the total amount of pixels flagged over the detector.). More in-depth analysis on a per-pixel basis, thus showing the location of issues, was carried out. However, the results obtained so far are fairly uniform for the entire detector. Interested readers are referred to the monitoring website of the SWIR module [4].

### 4.1   Background

Figure 18 and 19 show the detector median for the dark flux and offset from April 30th 2018 to April 30th 2019. Both are extremely stable. Even on a per-pixel basis, variations are very small, on scales of a few electrons (or electrons per second in the case of dark flux). Larger-scale variations seen during the monitoring have been exclusively caused by irregularities in the thermal controls, caused by orbit maneuvers. Data during such maneuvers are omitted in the data shown in Figures 18 and 19.

---

[4]http://www.sron.nl/tropomi-swir-monitoring/



## 4.2 Noise

Figure 20 shows the median noise of of the SWIR detector as a function of time from April 30th for a year. There is some variation, but most is much smaller than the typical spread of the in-flight noise seen over the detector.

## 4.3 Detector Pixel Quality and Radiation impacts

Radiation impacts will gradually degrade the detector by causing pixels to become too noisy for retrieval or damage them to such a degree they stop functioning. Most of these impacts occur in the South Atlantic Anomaly.

Figure 21 shows the number of detector pixels flagged as bad or dead from March 2018 to April 2019. Over this period, ~200 detector pixels had their quality value drop to below 0.8 and ~30 to below 0.1. A linear fit through all orbits gives a loss of 42 detector pixels per 1,000 orbits in the category bad and 6 detector pixels per 1,000 orbits in the category dead. When

compared to the total amount of total pixels (250,000), current estimates show that less than an additional 0.6% will be bad or dead at the end of the envisioned 7 years lifetime of TROPOMI. It is good to note that this assumes detector pixels are lost at the - currently observed - linear rate of 0.1% per year. However, if detector pixels are lost due to cumulative cosmic ray impacts, the rate likely will become non-linear at later stages during the lifetime. More in-depth analysis (i.e. using data from longer operational timescales) of the effects of cosmic ray impacts is warranted and planned for future work.

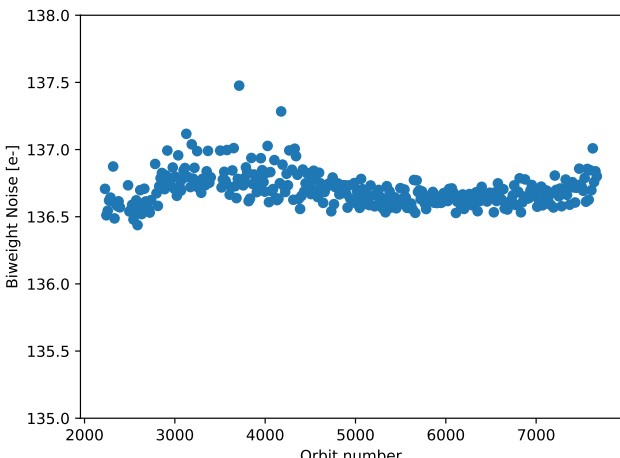

**Figure 20.** Median noise as a function of time. Large instabilities have been removed from this trend. Note that the median noise is insensitive for cosmic ray impacts.

## 4.4 Diffusers

Figure 22 shows the normalized response of the daily (which uses the main diffuser) or weekly (which uses the backup diffuser) solar irradiance measurements. The diffusers do not appear to degrade at the SWIR wavelengths. Note that the diffusers are

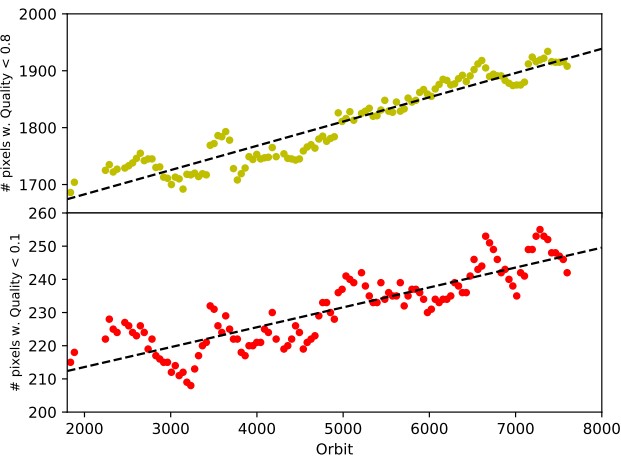

**Figure 21.** Number of bad (quality < 0.8, top image, yellow dots) and dead (quality <0.1, bottom image red dots) detector pixels since March 2018. A linear fit is shown with a black dashed line for each type. These have slopes of 42 and 6 detector pixels per 1,000 orbits for bad and dead detector pixels respectively.

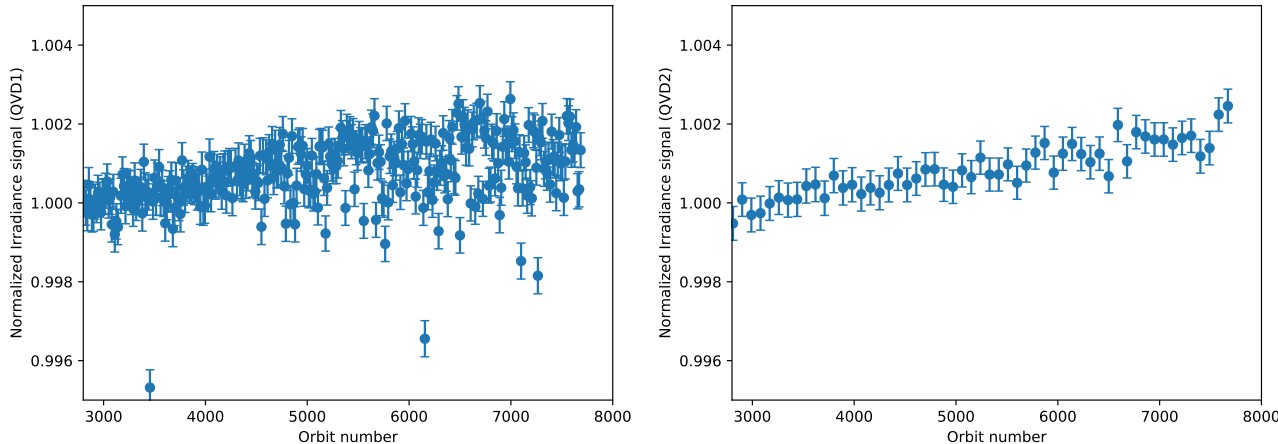

**Figure 22.** Median normalized solar irradiance signal as a function of time. Top is the daily solar irradiance measurements, performed using the main diffuser QVD1. Bottom is the weekly solar irradiance measurement, performed using the backup diffuser QVD2.

used for all four channels simultaneously. However, a long-term variance is seen in both diffusers, with the diffusers apparently becoming more effective. This is hypothesized due to an uncalibrated factor in the relative irradiance. However, a change in reflectivity of the diffuser cannot be ruled out. Further study is required. As it is small, it has no observable effect on L2 products.





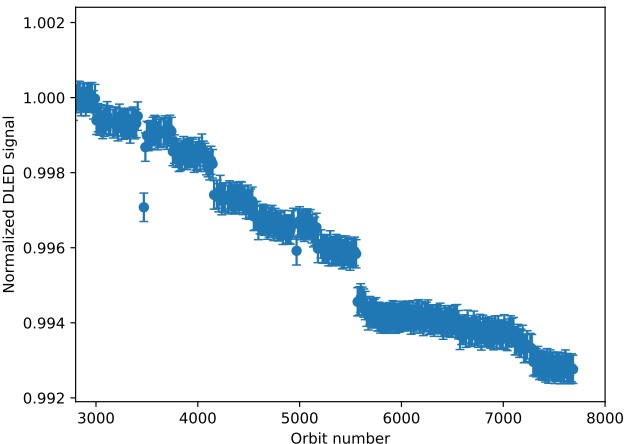

**Figure 23.** Median Normalized DLED signal as a function of time.

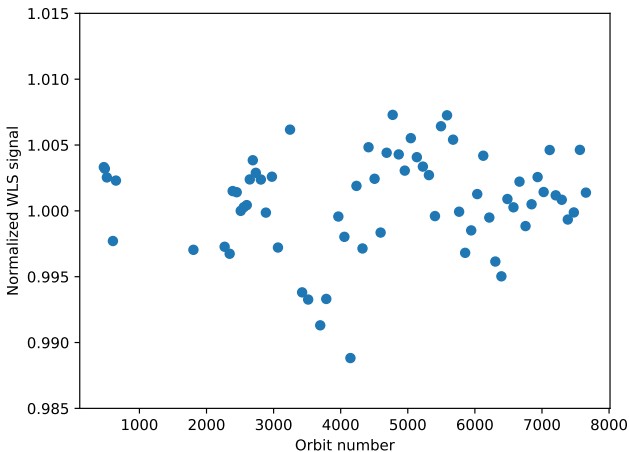

**Figure 24.** Median Normalized WLS signal as a function of time.

## 4.5 Stability of on-board calibration sources

Figures 23 and 24 show the normalized response of the DLED and WLS detector signals during nominal operations. The DLED signal is degrading at a rate ∼0.8 % per year. Given the increase of the solar irradiance signals as seen in Figure 22 it may be concluded that the DLED itself is degrading and not the detector responsivity. However, more monitoring is required to confirm this hypothesis.

The WLS signal appears not to degrade. Note however that the accuracy of the WLS measurements is limited. The output of the WLS varies within ∼1% (+0.5%, -0.5%), compared to the reference measurement. In addition, the spread of the values





around the median of each measurement also varies from measurement to measurement. However, degradation at levels seen for the DLED can be ruled out. It thus confirms a DLED degradation.

If we assume the DLED to degrade linearly, and given the envisioned lifetime of TROPOMI of seven years, the DLED is expected to lose 5.6% of its power output as compared to the start of nominal operations.

5   ## 4.6   ISRF

The stability of the ISRF is checked every month for each of the five diode lasers. Figure 25 shows the normalized pixel response in orbit 5396 as compared to orbit 1667. Both measurements use identical settings. The difference is of the same order as seen in the comparison with the on-ground measurements, reported earlier in section 3.6. Monthly comparison reveal residuals of at most 0.2%. These residuals vary from measurements to measurements due to the speckles on the diffuser.

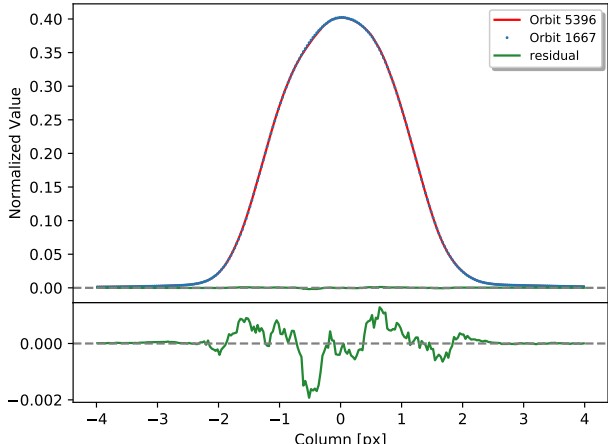

**Figure 25.** Comparison between the ISRF measurements taken during the commissioning phase and in nominal operations, both without an oscillating diffuser. *Top plot*: Normalized pixel response in orbits 5396 (red) and 1667 (blue). In green, the differences are plotted. *Bottom plot* row: Zoom of the residuals.

10   ## 4.7   Straylight

Stray-light monitoring is done once a month using diode laser SLS-1. Figure 26 reaffirms the conclusions and trend seen during the commissioning phase. Stray-light is found to be very stable, with the amount of total stray-light seen as a response to a line source to be ∼2.9%.

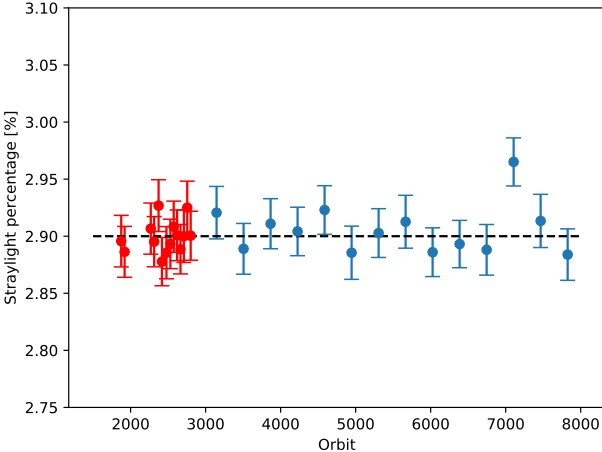

**Figure 26.** Results of all stray-light measurements with SLS-1 during the the commissioning phase (red, see table 4) and nominal operations (blue) phases.

## 5 Conclusions

From the results as presented in section 3 of this paper, it can be concluded that the SWIR module did not change between the on-ground calibration campaign and its first operations in space. This holds for all aspects: offset, dark flux, detector noise, transmission, stray-light and ISRF. The results of the first year of nominal operations, as presented in section 4 of this paper,
show that the SWIR module is very stable indeed on all aspects of the monitoring program. During the few orbit maneuvers that were needed to avoid collisions or to maintain formation flying with Suomi NPP, orientation of the satellite was lost resulting in non-nominal temperatures on board. Recovery to nominal temperatures was found to take hours for the SWIR detector and up to days for the SWIR spectrometer. During this time, small deviations from the regular CKD may occur. This is flagged in the data processer. The amount of pixels that have been lost so far is negligible (about 60 over 5 months), and given the current
rate, less than 0.6% of the total amount of pixels will be lost over the envisioned operational time of 7 years, assuming a linear rate (which may not be true in case of accumulated radiation damage). With the condition of the TROPOMI-SWIR module as it is now, a very stable operational period is foreseen with ample changes needed on the processer and on the calibration key data, yielding good quality Earth radiances to be used for accurate trace gas retrieval.

*Data availability.* The results shown in this paper were derived using the calibration data obtained from calibration measurements on-ground
and in-flight of Sentinel-5p. All data can be found in graphical form through the links describing the Calibration and validation activities found at https://sentinel.esa.int/web/sentinel/technical-guides/sentinel-5p/calibration . Specific data is available on request.

Atmospheric
*Competing interests.* There are no known competing interests

*Author contributions.* The bulk of the work was done by TvK, RvH and PT. RH was responsible for much of the analysis code, which was reviewed by PT en TvK. Analysis was equally shared by these three authors. IA was the interaction with other TROPOMI specialist teams. RH supervised the team up to the end of the commissioning phase.

5 *Acknowledgements.* This paper contains Copernicus Sentinel data. This research is funded by the TROPOMI national program from the Netherlands Space Office (NSO).



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
