# Peer review of "In-flight calibration and monitoring of the TROPOMI-SWIR module"

_Atmospheric Measurement Techniques, 2019_

## Referee Comment (RC1) · Anonymous Referee #1 · 23 Aug 2019

General comments:

The paper gives a good overview of the in-flight calibration and monitoring of the SWIR module and it is good to see that the SWIR module of TROPOMI is that stable over time.

Specific comments:

Page 2, line 7: 'dedicated the out-gassing the instrument' Is it not more likely for preventing contamination of the instrument from e.g. the platform, multi-layer insulation mainly water outgassing by making sure the instrument - especially if cooled SWIR - detectors do not act as a cold traps and get contaminated? Usually much care is taken with the instruments during integration, assembly and testing under cleanroom

environments that the outgassing of the instrument itself is minimised. Suggestion to rephrase.

Page 2, line 12: 'During nominal operations'. Not only during nominal operations, but the whole life time, even on-ground instruments should be monitored. Suggestion to rephrase.

Page 2, line 18: The word 'dark-flux' is used here and throughout the document, isn't it rather dark current? 'CKDs for . . .. were also derived on-ground.' Since some CKDs may also be derived on board, and may be updated, is the 'also' meaning also possible updated in flight? Suggestion to rephrase to make the statement more explicit and add also which CKDs are updated in flight.

Page 2, line 19: 'Signals of the sun as seen over the two diffuser' Please refer to paper with instrument design and/or add sketch of light path via diffuser for better understanding for the readers. 'internal lamps' Please explain which kind of lamps, refer to instrument design and add sketch of light paths.

Page 3, line 4: Isn't there also a CLED in the SWIR path? see e.g. Kleipool [2018] calibration unit description. Please add and shortly explain the light path.

Page 3, line 14: 'due to operational restrictions' this part is not understood, can you please detail what the restrictions are or why they are there or refer to another publication.

Page 3, line 18: according to Hees et al. [2018] this was already planned "However, as the diffuser mechanism is a life-limited item, only during the on-ground calibration campaign and during the in-flight commissioning phase, measurements will be performed with a moving diffuser." The way it is written now, it seems as if it was a later decision during operation. Suggestion to rephrase according to Hees et al. [2018].

Page 3, line 29: 'digitized typically with 12, 000' Is this binned on non-binned?

Page 4, line 1: 'Solar irradiance or signal from the on-board lights' this incl. straylight.

Page 4, line 4: 'amount of light lost' suggestion to change to degrading due to light loss, contamination

Page 6, figure 2.: The unit used here is Spectral Photon Radiance, why not use spectral radiance unit in [W m-3 sr-1]? Is this the unit used for the L1b products in the SWIR? As first image Iraq is shown, was there a special reason to select Basra? Why not e.g. another big city? or volcano?

Page 7, line 7: 'in form of blue bands' these blue bands in Fig.4 have negative numbers, is there less light in the background measurement with the absorption lines than during the on-ground measurements? was the temperature the same? Please detail in the text.

Page 8, figure 4.: Suggestion to add also the on-ground result as graph for better understanding.

Page 8, table 2: It seems difficult to compare FMM open with closed, since there are more than 1500 orbits difference, is it possible to compare open/ closed effects closer in time together? Also to have similar thermal conditions since the seasons are completely different. Otherwise there may be too many effects intermingled.

Page 9, figure 6.: Why is the uncertainty varying? Are e.g. different temperatures measured during these 6 weeks? Or are different amounts of measurements taken? please add text.

Page 10, line 10: 'clearly absent when the FMM is closed' Is then the conclusion to only measure background with the FMM closed? In case this is the conclusion, has the operational manual been adapted?

Page 10, line 24/25: 'no correction is applied for the SAA. . .' is there any conclusion about this? Currently it may be interpreted that the author would prefer to have a correction, but it might also just be a fact, that the SAA is flagged as in many other processors.
Page 12, line 4/5: 'Note that there appears a small systematic difference between the ADCs.' Detail what is meant here, which different ADCs, where is it visible , has it to do with the positive numbers in figure 7 up to column 500 and negative numbers from 500 to 1000?

Page 12, line 14: 'should not be used for retrieval of CO or CH4' this could be flagged, in case it is flagged and the processor is accounting for it accordingly suggestion to change to "is not used for retrieval of CO and CH4"

Page 12, line 17/18: 'Therefore, calibration measurements to determine noise levels should also be executed with the FMM closed' see previous comment: is this now operationally implemented? what are the issues with mission budget related to amount of movements of the FMM?

Page 12, line 23: 'cosmic ray impacts' aren't these flagged and can then be excluded from the processing?

Page 13, line 7: 'This includes pixels outside of the effective area, which are no illuminated.' But this does not mean they are not functional, they are/could still be used. Suggestion to reword.

Page 14, figure 10: 'similarity' why should these be the same, this is not understood? One is a difference between two methods on the same data, while the other is a difference in time and pre/post launch? Please detail.

Page 14, figure 11: It seems as if dead pixels came alive again over time, are these not rather 'pop-corn' pixels, thus in a way sometimes bad, sometimes good? Please consider renaming the dead pixels which become alive.

Page 15, table 3.: See also comment above. How can there be less dead pixels after launch than on-ground? or are these to be seen as additional number of bad/ dead quality pixels? Please detail.

Page 15, line 6/7: 'Although monitoring....' Why can it only be approximated? Please

detail.

Page 15, line 8/9: 'the calibration sources. . ...and/or diffusers.' and any other optical elements in the optical path incl. the video chain can be degraded. Please amend the text.

Page 15, line 20: 'hypothesized this is the . . ..grating.' This is not understood, since the DLED is in direct proximity of the detector and it was understood the light path from the DLED is not via the grating. Can it not be some kind of etalloning via the protective glass of the MCT detector, and a layer which might have been on it? Or is the CLED meant here? Please consider changing text.

Page 16, figure 12.: Please detail, e.g. top DLED 2515/907, bottom DLED2515/2707. The difference in the bottom is basically 1, thus no significant change measurable in about 200 orbits, is this understanding correct?

Page 16, line 7/8: 'less stringent stability limits of the WLS system. . ..due to the DLED.' Is it understood correctly, though the WLS with less stringent stability shows with the resulting expected larger error bars the same range as the DLED measurements? Please clarify.

Page 17, figure 14.: Would it be possible to show the differences by normalising to one measurement instead of plotting them on top of each other to better see the differences?

Page 18, line 1/2: See above, please plot also the ratios normalised to one measurement (similar as shown for the ISRF e.g. figure 15) to better support this sentence.

Page 23, figure 21.: The data seem to include 'pop-corn' pixels since number also decreasing again over time. See also comment above and consider changing the name 'dead pixels' for those who come alive again.

Page 23, figure 23.: Suggestion to check with the SLS via diffuser if the same trend is visible of increasing signal. After all 2018 was solar minimum.

Page 23, line 1/2: 'diffusers apparently becoming more effective' In this case this should then also be visible with the SLS via diffuser measurements over time. Has this been observed? Please detail in text/ with graphs.

Page 24, figure 23.: Around orbit 3500 is one significant outlier, is something visible from the housekeeping data to explain this outlier looking at the telemetry? In addition can the jump around orbit 5500 be explained?

Page 24, figure 24.: While most figures show error bars, they seem missing in this figure. Can they be added?

Page 24, line 2: 'a rate ∼0.8 % per year' From the housekeeping data, is in the telemetry the voltage/ amps of the LED given? Are these values stable over time? Can this be checked and added to the text.

Page: 26, figure 26.: Can the outlier with almost 3% around orbit 7500 be explained?

Page 26, line 9: '(about 60 over 5 months) ' What is the reason the choose here 5 months and not either since launch or during nominal operation?

Technical corrections general:

Figures are often in previous sections or next sections from their textual descriptions. For easier readability would it be possible to place them in the same section as where they are described?

Abbreviations should be minimised and at least the first time of appearance be fully written.

Page: 1 Line 1: short-wave infrared (SWIR) tropospheric monitoring instrument (TROPOMI)

Line 3: instrument spectral response function (ISRF)

Line 5: Change 'eclips side' to 'eclipse side'

Line 7: 'with little to no degradation' instead suggestion to provide values, e.g. smaller than <x>%

Line 16: ultra-violet, visible and near-infrared (UVN)

Line 19: 'TROPOMI will produce' change to 'TROPOMI produces'

Line 20: Change 'yeilding' to 'yielding'

Page: 2 Line 10: 'the E1 phase' change to 'phase E1'

Line 12: 'the E2 phase' change to 'phase E2'

Line 14: 'planet each day' change to 'planet Earth each day'

Line 15/16: and...and... Is used, suggestion to use a comma and then only one 'and'.

Line 33: 'radiance' change to 'spectral radiance'

Page: 3 Line 1 'irradiance' change to 'spectral irradiance'

Line 4: Add detector LED after DLED similar to the other two sources.

Line 17: 'oscillation mode' This sounds as if there is a motor installed on the diffuser mounts? Please refer to a paper with the instrument design, where this is described in more detail or suggestion to add text.

Line 26: 'each pixel is read out individually' does this imply it is a CMOS detector? Please add a little more detail about the detector and possibly refer to another publication.

Page: 4 Table 1.: PRNU - pixel to pixel non-uniformity

Line 10: Add '.' after 'complex correction algorithm.'

Line 4/5: 'Straylight is defined as any outside signal that does not follow the intended path onto the detector and is thus not part of the useful signal' The 'outside signal' may be confusing, since straylight may be ghosts, in-band straylight, out of field, out of

spectral band etc. straylight. Suggestion to rephrase for better understanding for the readers.

Page: 5 Figure 1.: Last part of flow, 'Useful radiance' : Radiance assumes already processed to spectral radiance units, with the help of CKDs, is this meant here? Possibly rephrase or add steps for generating spectral radiances.

Line 4: 'dark-flux' possibly other naming? Are dark current correction meant here? Flux maybe confusing.

Page: 6 Line 4: 'November 2019', is November 2017 meant here?

Page: 7 Line 4: 'dark flux' see previous comments.

Page: 8 Line 2: 'thedifferent' change to 'the different'

Table 2: 'On-ground diff.' unit missing. Please add

Page: 9 Figure 5.: 'during the commissioning phase . . .' during nominal operation, since you stated before "Nominal operations started at orbit number 2818." Please change text.

Figure 6.: 'commissioning phase' according to text above this is already nominal operational phase. Please change wording.

Page: 10 Line 18/19: 'thermal variations as a function of the orbital phase' suggestion to show a graph over the 1 1/2 year time with the thermal fluctuations from the housekeeping telemetry of the SWIR instrument.

Line 30: 'if the data taken in the SAA are excluded from the analysis' suggestion to add that this can be done due to the SAA flagging.

Line 31/32, last sentence: before a statement says they are different from Hoogeveen [2013], seems contradicting : "The amount of dark flux detected differs from the measured value reported in Hoogeveen et al. (2013). " Please bring in line, or detail.

Page: 11 Figure 7.: nominal operation phase, see comment above.

Figure 8: nominal operation phase, see comment above.

Page: 12 Line 5: ADC analogue digital converter.

Line 6: 'acceptable' suggestion to change word, like this it can be interpreted as 'unacceptable', since stated to be below acceptable levels.

Page: 13 Line 8: 'impacts or other hardware degradation' suggestion to eliminate the 'other' and change to 'impacts or hardware degradation'

Page: 14 Figure 11.: Please add dead (bottom red dots) and bad (top green dots).

Page: 15 Line 15: Add '.' after sentence.

Page: 16 Line 7: 'end of E1' change to end of phase E1' Is the WLS also in orbit 2515 similar to the DLED? Please provide orbit.

Page 18, line 15: Add 'phase E1' instead of 'E1'

Page: 19 Line 2: 'but conclusions apply to results obtained' it is assumed 'but conclusions also apply. . .' if correct please change text.

Page: 20 Line 1/2: Please in one line 'The' with a space in front.

Page: 21 Line 15: 'from April 30th. . ..' isn't this already from an earlier date, since starting already at orbit 1800 or so. Please change.

Line 17/18: 'Larger-scale variations seen. . ..manoeuvres.' Please provide orbits when this happened linked to the figures.

Page: 23 Figure 22.: Instead of 'Top' change to 'left' and 'bottom' to 'right'.

Page: 25 Line 4: '5.6 %' might it be better to just state 6%, since with the kind of step degradations seen in figure 24 the number 5.6 may suggest a higher accuracy of the assumed linear prediction of degradation? Consider changing.

---

## Referee Comment (RC2) · Anonymous Referee #2 · 10 Sep 2019

The comment was uploaded in the form of a supplement:
https://www.atmos-meas-tech-discuss.net/amt-2019-270/amt-2019-270-RC2-supplement.pdf

---

## Referee Comment (RC3) · Anonymous Referee #3 · 16 Sep 2019

This paper covers the first year or so of on-board calibration for the SWIR module for TROPOMI in a very straightforward, systematic way. There are few surprises, which is always a good thing when it comes to spaceborne remote sensors. The techniques discussed have been presented elsewhere in the literature (and hence are not novel), but the documentation of the TROPOMI instrument status at this stage is in itself a useful and important work. I recommend publication after minor corrections.

Figure 4: There's no mention of the linear features in the radiances at indices 400 and 500 - are these dark flux or something else? Is there a mask on the detector for the top and bottom rows? The results look very different compared to the middle of the detector.

Page 8, Line 3-4: Did they not have any thermal measurements for an actual compari-

[Figure]

son? This difference may be important.

Figure 7: There's a pretty distinct discontinuity in the offset at the middle of the detector, but I don't see any discussion of this in the text.

Figure 8: There seems to be a linear trend in the median offset , but that could be an artifact of the statistics rather than a real thing. Which is it?

Figure 11: How is it possible that the number of bad/dead pixels decreases with time? Did your criteria change?

Figure 17 shows that this could be a 1% error, which is significant for trace gas retrievals.

Page 17, Line 5-6: This is really only a measurement of a small area of the bandpass. Is there any reason to assume that this is representative of all of the other wavelengths as well? Please note that I realize this is a heroic effort to track stray light.

What is the difference between Figures 16 and 17?

Page 26, Line 2: I disagree, the offset image in Figure 7 shows a pretty clear difference.

---

## Short Comment (SC1) · 17 Sep 2019

Comments appear to be empty. Was this the intention?

---

## Referee Comment (RC4) · Anonymous Referee #2 · 18 Sep 2019

Dear author,

the comment can be found in the attachment. Yo need to click on the floppy disk symbol with the green S on it.

---

## Referee Comment (RC5) · Anonymous Referee #4 · 1 Oct 2019

General Comments: As most of the issues in this paper were already covered by the comments of anonymous referees #1,#2 and #3, I will mostly focus on issues not sufficiently covered by the other reviewers.

I agree with referee #3, that the techniques discussed in this manuscript have already been presented elsewhere in the literature and therefore are not novel, but the documentation of the TROPOMI instrument status is in itself a useful and important work. Therefore I also recommend publication after minor corrections.

As however the manuscript seems largely to be extracted from a status report, I would suggest changing the title of the manuscript to "Technical Note: In-flight calibration and monitoring of the TROPOMI-SWIR module".

[Figure]

I also recommend adding consistent captions to all figures (e.g. figure 6: [e-/sec] is missing, figure 8: electronic offset is reported in V, rather than in [e-] and therefore magnitude and variation of the offset could not be compared quickly to the magnitude of the DC variation).

I also wonder, why the eclipse side of the Earth was assumed pre-launch as dark, as black body radiance at $\sim$ 293K is non zero in the spectral range $> \sim$ 1750 nm. Therefore it is not astonishing, that the TROPOMI-SWIR module detects thermal radiation of the earth with the FFM open.

When discussing the DC, it should be mentioned that the DC consists of two primary components, Detector DC and DC introduced by the thermal radiation of the instruments optical bench itself.

Reviewer #3 also asks if there is a mask implemented on the detector. Some literature research (i.e. Paul et al., Characterization and correction of stray light in TROPOMI-SWIR) reveals, that "In the grey area, the light is blocked by the entrance slit of the spectrometer (top and bottom) or a shield at the detector (left and right)", see caption "Figure 2" of this manuscript. As shielded rows represent an excellent possibility to disentangle detector dark current from ambient dark current (produced from the optical bench), I would strictly recommend expanding the DC analysis including that dataset, given that such data is available. Such data could also reveal potential problems with dark signal shifts known to exist when illuminating larger parts of MCT-SWIR detectors, see for instance: Chapman et. al, "Spectral and Radiometric Calibration of the Next Generation Airborne Visible Infrared Spectrometer (AVIRIS-NG)", and according explanation of pedestrial shift.

As mentioned by Paul et al. light on the SWIR detector is also blocked by the entrance slit on top and bottom of the spectrometer. Typically, such measurement could be used to assess the In-Band stray light level of the instrument. Therefore I recommend to add an according analysis, if such data is available for the TROPOMI SWIR spectrometer

module.

Can you furthermore give more explanation on the possible cause of the fringes observed in Fig. 12, top (DLED measurements) ?

In addition I would recommend adding sketches of the optical path for the different measurement configuration as these will ease up to follow the different optical configurations mentioned in the manuscript.

―――――――――――――――

---

## Author Comment (AC1) · 1 Nov 2019

*Dear Mrs./Mr., We would like to thank the referee for their time and effort in reading and reviewing our paper with constructive comments. Please find our answers to your comments below in italics.*

General comments: The paper gives a good overview of the in-flight calibration and monitoring of the SWIR module and it is good to see that the SWIR module of TROPOMI is that stable over time.

*Thank you*

Specific comments:

[Figure]

Page 2, line 7: 'dedicated the out-gassing the instrument' Is it not more likely for pre-venting contamination of the instrument from e.g. the platform, multi-layer insulation mainly water outgassing by making sure the instrument - especially if cooled SWIR - detectors do not act as a cold traps and get contaminated? Usually much care is taken with the instruments during integration, assembly and testing under cleanroom environments that the outgassing of the instrument itself is minimised. Suggestion to rephrase.

*We agree, rephrased.*

Page 2, line 12: 'During nominal operations'. Not only during nominal operations, but the whole life time, even on-ground instruments should be monitored. Suggestion to rephrase.

*We agree, rephrased.*

Page 2, line 18: The word 'dark-flux' is used here and throughout the document, isn't it rather dark current? 'CKDs for . . ... were also derived on-ground.' Since some CKDs may also be derived on board, and may be updated, is the 'also' meaning also possible updated in flight? Suggestion to rephrase to make the statement more explicit and add also which CKDs are updated in flight.

*We agree that the difference in terms between dark flux and dark current is rather con-fusing. This is caused by a rather complicated issue. Dark current is defined as the current produced by the detector at its operational temperature. However, at the SWIR wavelengths, the measured dark current is always an addition of the true dark current (produced by the detector itself) and the signal from an external thermal background (i.e. anything the detector looks at in the dark). Even if something like a baffle is cov-ering the detector, this will produce a signal equal to the Planck curve at the baffle temperature in addition to the true dark current. Internally we defined and used dark flux to distinguish it from a true instrumental dark current. During the instrument de-velopment phase this proved to be very valuable. The 'also' was in reference to the*

*straylight and ISRF CKDs. All CKDs currently used are derived on-ground. Rephrased slightly for clarity. We have the capability to derive and update some of the CKDs in-flight (dark flux, offset, Quality mask, noise), but have so far not seen a reason for this. This capability has not been clearly described, but is now included in the start of section 2.*

Page 2, line 19: 'Signals of the sun as seen over the two diffuser' Please refer to paper with instrument design and/or add sketch of light path via diffuser for better understanding for the readers. 'internal lamps' Please explain which kind of lamps, refer to instrument design and add sketch of light paths.

*The instrument design, which includes the light path description and its requirements on the solar irradiance path via multiple diffusers fall under commercial control of Airbus and TNO and not to SRON or any of the authors. They were sadly not published. We have included an overview figure with the location of the on-board light sources with the light paths of the SLS and WLS. The location of the CLED is shown, but as it is not usable, no light paths are indicated. The DLED is located inside the module next to the detector.*

Page 3, line 4: Isn't there also a CLED in the SWIR path? see e.g. Kleipool [2018] calibration unit description. Please add and shortly explain the light path.

*There is, but the emission properties of the CLED (i.e. the amount of photons produced) do not extend into the SWIR wavelength range. This was checked during on-ground calibration. We added a footnote to explain this.*

Page 3, line 14: 'due to operational restrictions' this part is not understood, can you please detail what the restrictions are or why they are there or refer to another publication.

*This has two reasons. First, the amount of time designated for calibration with one of the on-board lights (DLED, WLS or lasers) during the eclipse side of an orbit is only*

*a few minutes. This has been rephrased and referred to van Hees et al., 2018, which detail the operational restrictions.*

Page 3, line 18: according to Hees et al. [2018] this was already planned "However, as the diffuser mechanism is a life-limited item, only during the on-ground calibration campaign and during the in-flight commissioning phase, measurements will be performed with a moving diffuser." The way it is written now, it seems as if it was a later decision during operation. Suggestion to rephrase according to Hees et al. [2018].

*Rephrased*

Page 3, line 29: 'digitized typically with 12, 000' Is this binned on non-binned?

*unbinned*

Page 4, line 1: 'Solar irradiance or signal from the on-board lights' this incl. straylight.

*Agree. Rephrased*

Page 4, line 4: 'amount of light lost' suggestion to change to degrading due to light loss, contamination

*Rephrased*

Page 6, figure 2.: The unit used here is Spectral Photon Radiance, why not use spectral radiance unit in [W m-3 sr-1]? Is this the unit used for the L1b products in the SWIR? As first image Iraq is shown, was there a special reason to select Basra? Why not e.g. another big city? or volcano?

*Spectral Photon radiance is indeed the unit used in L1b products and adopted here for consistency with the L1b released product. Basra was selected as it is the clearest example of the contamination. The signal also originates not from Basra itself but from the many oil drilling sites near the city. These burn off excess gas at night, and the fire is the origin of the signal. We do not see the city itself. Signals of big cities are typically not seen at the eclipse side. Other sources seen are natural wildfires. Volcano's have*

*not detected., but in the limited available data (background measurements with the FMM open during the E1 phase) we have not actively looked for them.*

Page 7, line 7: 'in form of blue bands' these blue bands in Fig.4 have negative numbers, is there less light in the background measurement with the absorption lines than during the on-ground measurements? was the temperature the same? Please detail in the text.

*Yes, During the on-ground measurements there was no air between the light source and detector.*

Page 8, figure 4.: Suggestion to add also the on-ground result as graph for better understanding.

*Respectfully disagree: given the very small relative differences (<0.2%), plotting the on-ground results on absolute scale is not going to be very illustrative.*

Page 8, table 2: It seems difficult to compare FMM open with closed, since there are more than 1500 orbits difference, is it possible to compare open/ closed effects closer in time together? Also to have similar thermal conditions since the seasons are completely different. Otherwise there may be too many effects intermingled.

*In answer to your first question: sadly no. Due to the schedule of the E1 period, open and closed comparison will always have a minimum 1200 orbits in between (orbit 2240 is about the earliest where enough useful measurements were obtained. see Fig. 6). During orbit 1100-2200 new calibration measurements for the relative irradiance were obtained. As such, closer in time can only be done by about 25%, which is not significant. However, it is our opinion given the results of the closed FMM monitoring (see Fig. 6) and the very low variance seen there, that any closed FMM measurement can be compared to any open FMM measurement.*
*On your second question: Thermal conditions of the SWIR instrument and all components in the optical path are not influenced by the seasons. The thermal conditions are*

[Figure]

*nearly identical between all of the presented orbits.*

Page 9, figure 6.: Why is the uncertainty varying? Are e.g. different temperatures measured during these 6 weeks? Or are different amounts of measurements taken? please add text.

*Text added. It is the amount of measurements.*

Page 10, line 10: 'clearly absent when the FMM is closed' Is then the conclusion to only measure background with the FMM closed? In case this is the conclusion, has the operational manual been adapted?

*Text added. And the answers are yes, and yes.*

Page 10, line 24/25: 'no correction is applied for the SAA. . .' is there any conclusion about this? Currently it may be interpreted that the author would prefer to have a correction, but it might also just be a fact, that the SAA is flagged as in many other processors.

*Rephrased.*

Page 12, line 4/5: 'Note that there appears a small systematic difference between the ADCs.' Detail what is meant here, which different ADCs, where is it visible, has it to do with the positive numbers in figure 7 up to column 500 and negative numbers from 500 to 1000?

*Yes this has been rephrased.*

Page 12, line 14: 'should not be used for retrieval of CO or CH4' this could be flagged, in case it is flagged and the processor is accounting for it accordingly suggestion to change to "is not used for retrieval of CO and CH4"

*Done*

Page 12, line 17/18: 'Therefore, calibration measurements to determine noise levels

should also be executed with the FMM closed' see previous comment: is this now operationally implemented? what are the issues with mission budget related to amount of movements of the FMM?

*Correct. This was implemented by the team responsible for operations. FMM closing is indeed not done every orbit. However, since it was already planned for every other orbit, no additional FMM movements had to be scheduled so there is no effect on the number of movements of the FMM wrt originally planned operations schedule. This has little effect on the calibration quality.*

Page 12, line 23: 'cosmic ray impacts' aren't these flagged and can then be excluded from the processing?

*These are indeed flagged. However, flagging is done much later in the processing chain than typically processing level for noise measurements. In addition, the noise must also be able to distinguish cosmic ray impacts from actual noisy pixels. False positive/negatives or too rigorous flagging may obfuscate noisy pixels. Therefor a choice was made not to rely on flagging. Noisy pixels remain noisy for all measured frames, while cosmic ray impacts do not.*

Page 13, line 7: 'This includes pixels outside of the effective area, which are no illuminated.' But this does not mean they are not functional, they are/could still be used. Suggestion to reword.

*Agreed. Rephrased.*

Page 14, figure 10: 'similarity' why should these be the same, this is not understood? One is a difference between two methods on the same data, while the other is a difference in time and pre/post launch? Please detail.

*The difference between the on-ground and in-flight mean is dominated by impacts of cosmic rays. During on-ground calibration there are no cosmic ray hits, while in-flight measurements are riddled with them. The bi-weight spread method effectively removes*

*the influence of the cosmic rays on the noise calculation. As such these figures look remarkably similar. This was detailed on page 12.*

Page 14, figure 11: It seems as if dead pixels came alive again over time, are these not rather 'pop-corn' pixels, thus in a way sometimes bad, sometimes good? Please consider renaming the dead pixels which become alive.

*In an initial classification we adopted many more categories. This very quickly became very too complicated with many different categories if we wanted to become very consistent ("outside illuminated area" , "truly dead" – 0.0 remaining 0.0. "recoverable dead" – 0.0 but can recover. "Truly very bad" ($>$0.0 and $<$0.1 remaining there), etc. etc.). In addition to this, pop-corn pixels can exist in various flavors. First there are pop-corn pixels that are bad/good every few orbits (skirting the relatively arbitrary divide of 0.8). Second are pop-corn pixels that only change the noise behavior after annealing (i.e. heating of the array, which typically happens unplanned during anomalies). Given the complexity of such a categorization, there is the possibility any pixel may still change category (e.g. a pixel that has always been 'truly dead' may still turn into a pop-corn pixel) and the relatively small number of pixels ($\sim$ 2,000 on $\sim$210,000 in the illuminated area) , a choice was made to simplify this to three categories ('dead', 'bad' and 'good'). We agree that the nomenclature of the 'dead' category can be a little misleading. As such we have added a footnote.*

Page 15, table 3.: See also comment above. How can there be less dead pixels after launch than on-ground? or are these to be seen as additional number of bad/ dead quality pixels? Please detail.

*See above. The 'dead' label is a category with quality $<$0.1 and 'dead' was chosen as a nomenclature. There are several effects in play. The most likely is the thermal environment, which is much more stable in-flight than achieved on-ground.*

Page 15, line 6/7: 'Although monitoring. . ..' Why can it only be approximated? Please detail.

*Because we do not have access to external lamps (outside of the irradiance). Therefor several components cannot be characterized (using an external light source other than the sun over the solar diffusers is not possible for instance).*

Page 15, line 8/9: 'the calibration sources. . ...and/or diffusers.' and any other optical elements in the optical path incl. the video chain can be degraded. Please amend the text.

*Done*

Page 15, line 20: 'hypothesized this is the ... grating.' This is not understood, since the DLED is in direct proximity of the detector and it was understood the light path from the DLED is not via the grating. Can it not be some kind of etalloning via the protective glass of the MCT detector, and a layer which might have been on it? Or is the CLED meant here? Please consider changing text.

*That is indeed a possibility. The hypothesis was not the grating itself, but the thermal balance (or lack thereof) in the instrument due to the very long relaxation times of the grating. However, after further looking into this, we can confirm that the hypothesis proposed by the referee is possible as well (either glass over the MCT detector, or a lens in the optical path to the DLED). We changed the text accordingly.*

Page 16, figure 12.: Please detail, e.g. top DLED 2515 - 907, bottom DLED 2515 - 2707. The difference in the bottom is basically 1, thus no significant change measurable in about 200 orbits, is this understanding correct?

*Done, and that is correct.*

Page 16, line 7/8: 'less stringent stability limits of the WLS system. . ..due to the DLED.' Is it understood correctly, though the WLS with less stringent stability shows with the resulting expected larger error bars the same range as the DLED measurements? Please clarify.

*That is correct, and we added this to the text.*

Page 17, figure 14.: Would it be possible to show the differences by normalising to one measurement instead of plotting them on top of each other to better see the differences?

*Respectfully, we disagree with the referee. A figure as proposed by the referee was made before submission, but resulted in a) an identical figure near the peak and b) a very confusing figure near the edges of the detector where very little diode laser light is detected. Given the necessary information of the absolute scales at various distances from the laser illuminated area, we decided on this version of the figure.*

Page 18, line 1/2: See above, please plot also the ratios normalised to one measurement (similar as shown for the ISRF e.g. figure 15) to better support this sentence.

*The changes shown are typical measurement errors (given in the uncertainty in the second column), which cannot be easily represented in a normalized version, see above.*

Page 23, figure 21.: The data seem to include 'pop-corn' pixels since number also decreasing again over time. See also comment above and consider changing the name 'dead pixels' for those who come alive again.

*See above answer*

Page 23, figure 23.: Suggestion to check with the SLS via diffuser if the same trend is visible of increasing signal. After all 2018 was solar minimum.

*This is a very constructive suggestion and is a possibility we considered ourselves. However, due to the life limited item (oscillating diffuser) and amount of heat introduced into the system, such measurements are not carried out during regular operations and thus cannot be easily verified. Alternatively, this hypothesis can be proven correct by monitoring the diffusers over much longer times (three to four years). Note that at shorter wavelengths (UV to VIS), a degradation has been confirmed in the diffuser coating. This may very well have propagated to the SWIR wavelengths at very low*

*levels. Again, either measurements with an oscillating diffuser (currently unavailable) or very long-term monitoring will provide an answer.*

Page 23, line 1/2: 'diffusers apparently becoming more effective' In this case this should then also be visible with the SLS via diffuser measurements over time. Has this been observed? Please detail in text/ with graphs.

*The SLS signal does not pass over the same diffuser. Instead it has its own diffuser (which is never illuminated by the solar signal)*

Page 24, figure 23.: Around orbit 3500 is one significant outlier, is something visible from the housekeeping data to explain this outlier looking at the telemetry? In addition can the jump around orbit 5500 be explained?

*The outlier in orbit 3500 was due to a space craft anomaly in the spacecraft. This has been added to the text. The jump around orbit 5500 cannot be traced to a specific event.*

Page 24, figure 24.: While most figures show error bars, they seem missing in this figure. Can they be added?

*The statistical WLS error bars are omitted as they are not representative due to the uncorrected non-linearity. This has been added to the text in 3.4.2*

Page 24, line 2: 'a rate 0.8 % per year' From the housekeeping data, is in the telemetry the voltage/ amps of the LED given? Are these values stable over time? Can this be checked and added to the text.

*See attached figure. Voltage of the LED is (nearly) constant over time. Has been added to the text*

Page: 26, figure 26.: Can the outlier with almost 3

*No, it cannot*

Page 26, line 9: '(about 60 over 5 months) ' What is the reason the choose here 5 months and not either since launch or during nominal operation?

*There are very little good reference measurements during the early parts of E1. We specifically chose to have the End of E1 to be the reference. The 5 months was a leftover from an earlier draft that used 5 months of data from the reference point. The current figure uses a full year of data from that reference point. That has been corrected.*

Technical corrections general:

*In general all technical corrections are adopted. Some comments are added to individual comments.*

Figures are often in previous sections or next sections from their textual descriptions. For easier readability would it be possible to place them in the same section as where they are described?

*This has been attempted, but is a limitation of the 1- or 2-column format*

Abbreviations should be minimised and at least the first time of appearance be fully written. Page: 1 Line 1: short-wave infrared (SWIR) tropospheric monitoring instrument (TROPOMI)
Line 3: instrument spectral response function (ISRF)
Line 5: Change 'eclips side' to 'eclipse side'
Line 7: 'with little to no degradation' instead suggestion to provide values, e.g. smaller than <x>%

*I do not think it is correct to express this into a single number as we include the detector, onboard lamps and grating. A single number can be misleading. The sentence has been rephrased.*

Line 16: ultra-violet, visible and near-infrared (UVN)

Line 19: 'TROPOMI will produce' change to 'TROPOMI produces'

Line 20: Change 'yeilding' to 'yielding'

Page: 2 Line 10: 'the E1 phase' change to 'phase E1'

Line 12: 'the E2 phase' change to 'phase E2'

Line 14: 'planet each day' change to 'planet Earth each day'

Line 15/16: and...and... Is used, suggestion to use a comma and then only one 'and'.

Line 33: 'radiance' change to 'spectral radiance'

Page: 3 Line 1 'irradiance' change to 'spectral irradiance'

Line 4: Add detector LED after DLED similar to the other two sources.

Line 17: 'oscillation mode' This sounds as if there is a motor installed on the diffuser mounts? Please refer to a paper with the instrument design, where this is described in more detail or suggestion to add text.

*See van Hees et al., 2018 for clearest explanation. Instrument design paper was not published and is property of industry (TNO and Airbus Defense and Space Netherlands). But yes, oscillating the mounts does require a motor.*

Line 26: 'each pixel is read out individually' does this imply it is a CMOS detector? Please add a little more detail about the detector and possibly refer to another publication.

*Yes it is. Sadly no publication exists clearly describing this.*

Page: 4 Table 1.: PRNU - pixel to pixel non-uniformity

Line 10: Add '.' after 'complex correction algorithm.'

Line 4/5: 'Straylight is defined as any outside signal that does not follow the intended path onto the detector and is thus not part of the useful signal' The 'outside signal' may be confusing, since straylight may be ghosts, in-band straylight, out of field, out of spectral band etc. straylight. Suggestion to rephrase for better understanding for the

readers.

Page: 5 Figure 1.: Last part of flow, 'Useful radiance' : Radiance assumes already processed to spectral radiance units, with the help of CKDs, is this meant here? Possibly rephrase or add steps for generating spectral radiances.

*es, this is meant here. Rephrased to indicate unit conversions also take place. These are not monitored, and as such omitted from this diagram.*

Line 4: 'dark-flux' possibly other naming? Are dark current correction meant here? Flux maybe confusing.
Page: 6 Line 4: 'November 2019', is November 2017 meant here?
Page: 7 Line 4: 'dark flux' see previous comments.
Page: 8 Line 2: 'thedifferent' change to 'the different'
Table 2: 'On-ground diff.' unit missing. Please add
Page: 9 Figure 5.: 'during the commissioning phase . . .' during nominal operation, since you stated before "Nominal operations started at orbit number 2818." Please change text.
Figure 6.: 'commissioning phase' according to text above this is already nominal operational phase. Please change wording.
Page: 10 Line 18/19: 'thermal variations as a function of the orbital phase' suggestion to show a graph over the 1 1/2 year time with the thermal fluctuations from the housekeeping telemetry of the SWIR instrument.

*The timescales here do not match in this comment.. Orbital phase is a changing parameter within a single orbit. A graph over 1,5 years of thermal fluctuations does not represent information on this. We added a definition for orbital phase for this.*

Line 30: 'if the data taken in the SAA are excluded from the analysis' suggestion to add that this can be done due to the SAA flagging.

*This could not be done at that stage due to the processor version. It was done by hand. Changed wording from 'if' to 'when'*

Line 31/32, last sentence: before a statement says they are different from Hoogeveen [2013], seems contradicting : "The amount of dark flux detected differs from the measured value reported in Hoogeveen et al. (2013). " Please bring in line, or detail.
Page: 11 Figure 7.: nominal operation phase, see comment above.
Figure 8: nominal operation phase, see comment above.

*These figures use data from end of E1.*

Page: 12 Line 5: ADC analogue digital converter.
Line 6: 'acceptable' suggestion to change word, like this it can be interpreted as 'unacceptable', since stated to be below acceptable levels.
Page: 13 Line 8: 'impacts or other hardware degradation' suggestion to eliminate the 'other' and change to 'impacts or hardware degradation'
Page: 14 Figure 11.: Please add dead (bottom red dots) and bad (top green dots).
Page: 15 Line 15: Add '.' after sentence.
Page: 16 Line 7: 'end of E1' change to end of phase E1' Is the WLS also in orbit 2515 similar to the DLED? Please provide orbit.

*It was orbit 2513.*

Page 18, line 15: Add 'phase E1' instead of 'E1'
Page: 19 Line 2: 'but conclusions apply to results obtained' it is assumed 'but conclusions also apply. . .' if correct please change text.
Page: 20 Line 1/2: Please in one line 'The' with a space in front.
Page: 21 Line 15: 'from April 30th. . ..' isn't this already from an earlier date, since

starting already at orbit 1800 or so. Please change.

*This is correct. It has been monitored even before orbit 1800, although the backgrounds were shown to be unusable. However, the references are taken in the week before April 30th. Text has been changed to reflect this.*

Line 17/18: 'Larger-scale variations seen. . ..manoeuvres.' Please provide orbits when this happened linked to the figures.

*Data is typically only affected by 1-3 orbits. This means that the plotted number use less data (i.e. 6 instead of 8 input orbits). Only during very significant events would it show up in the figure. The only one is during orbit 3500 when an FDIR anomaly took place.*

Page: 23 Figure 22.: Instead of 'Top' change to 'left' and 'bottom' to 'right'.

*In a two column version of the paper, top and bottom are correct.*

Page: 25 Line 4: '5.6%' might it be better to just state 6%, since with the kind of step degradations seen in figure 24 the number 5.6 may suggest a higher accuracy of the assumed linear prediction of degradation? Consider changing.

---

## Author Comment (AC2) · 1 Nov 2019

Author response

*We would like to thank the referee for their time and effort in reading and reviewing our paper with constructive comments. Please find our answers to your comments below in italics.*

General

*A word on the structuring: In an initial draft we had set up a different structure. However, we found that this did not enhance the readability and made for some confusion on some topics (most notably backgrounds and quality, but also the readability of the ISRF proved to be confusing). We are aware of duplication of some of the data in*

[Figure]

*some figures. But in our opinion this division improved the readability of the paper greatly and more clearly represented the work done over the first year, where a clear separation between the E1 and E2 phases are present.*

SC 1 Haven't smaller spatial pixels been implemented in the mean time? This might be worth mentioning

*This is correct, but the smaller pixels were implemented only after first submission. Text has been added.*

SC 2 "Derivations of the ISRF": Are the different derivation approaches for the ISRF reported in the cited article or is the plural form a typo? If there are multiple ways it might be good to write this more explicit.

*The derivations are meant to indicate those of the ISRF and of Straylight. Rephrased.*

SC 3 This sentence is a bit confusing and seems not very accurate. Please specify the internal sources and maybe re-phrase the sentence by starting with transmission changes.

*This section has been rephrased and clarified.*

SC4 It is worthwhile to mention the two internal solar diffusers here.

*This section has been rephrased and clarified.*

SC 5 "Background measurements": Later you introduce background measurement with an open and a closed folding mirror, it would be useful to already introduce the two different types here.

*The background measurements with the FMM open are the same as spectral radiance measurements at the night-side. Mentioned on page 2, line 32. Text has been added for clarity.*

SC6 Are SLS and WLS following the complete optical path or not? The two sentences

seem to contradict each other. What do you call the complete optical path? The path the radiance takes or the path the irradiance takes?

*This section has been rephrased and clarified. TROPOMI uses a wheel to create different optical paths. The optical path beyond the wheel is identical for the SLS/WLS, the radiance and irradiance.*

SC7 SLS passes over: Isn't the light reflected off the side of the solar diffuser? Or does it pass through it? Are both diffusers used with the SLS?

*This is a dedicated diffuser, not one of the solar diffusers.*

SC 8 Are all detector pixels illuminated and used for science?

*No, they are not. In total 960 columns (spectral dimension) and 215 rows (spatial dimension) can be, illuminated.*

SC 9 The terms static and dynamic are not explained. Are these terms relevant? If not, remove them.

*Rephrased. Text added for clarity.*

SC 10 So it doesn't depend on the exposure time but on co-addition if a correction is applied or not? The first sentence is confusing

*For TROPOMI-SWIR co-addition factor and exposure time are linked due to the master cycle time of either 1080 or 800 ms. But in effect yes, the memory correction is only applied if the data is not co-added.*

SC 11 The figure is specific to a certain correction? This is not clear from the sentence

*Text added for clarity.*

SC 12 Are these all processing steps for the SWIR module? There seems to be no non-linearity, memory effect or exposure time correction. Please adapt the caption to make clear it's a simplified scheme or complete the flow chart.

*The figure is indeed a summary/simplification. This has been added.*

SC13 The term "flux" is rather ambiguous and at least in physics associated with a current through an area. The term dark current might be less ambiguous.

*The term was changed to dark current. We attempted to avoid the term dark current as the 'dark flux/dark current' we measure in this paper is a combination of the inherent dark current (current from The detector itself) and the signal from the thermal surroundings.*

SC14 Is this also visible in a geo-plot of the Earth? A plot would be nice to illustrate your point.

*I am uncertain how a geo-plot would contribute to the improved uncertainty/spread with the FMM closed. This is only an effect of the number of inputs.*

SC15 The term dark flux is used for the background radiance that can be confusing

*Changed*

SC16 This sentence is not entirely clear: I assume it is the signal that varies over time? The on-ground value is constant (I assume), so there is no need to refer to the difference, or is there?

*The variation is dominated by the large-scale dark flux and not the individual sources, e.g. the fires. The location where the measurement is taken does vary due to the sun-synchronous orbit. Whether or not we include more land or ocean based measurements and the weather above these Sites influence it.*

SC17 Is the orbit from nominal operations with open or closed FMM? What is meant by On-ground Diff.? On-ground minus in-flight or the other way around?

*Nominal operations is with FMM closed. Difference is On-ground minus in-flight.*

SC18 "commissioning phase": shouldn't it be E2 phase according to the orbit range

you specify in Section 1?

*The E1 phase is meant by the term commissioning phase. That has been changed.*

SC19 Is the mapping between columns and spectral direction exact or is there a spectral smile? (Also Fig. 7)

*There is a small spectral smile, but this is neglible.*

SC20 Why is there a gap in the plot?

*No background measurements were taken during that time due to scheduled other calibration measurements.*

SC21 Is temperature data available? Can you confirm your assumption?

*The temperature data inflight is available (see www.sron.nl/tropomi-monitoring), however, the on-ground thermal stability data was found to vary much more significantly. This data is also not public, but we confirmed the thermal difference. Whether or not that is truly the difference cannot be easily confirmed as the scheduled thermal tests during the E1 phase were not executed.*

SC22 This is not very clear, is it the same as in 3.1.1 line 9? Why the requirement of 40

*This is technically the same requirement. However, with FMM open, background measurement can (and were) taken every orbit. To save the life-limited-usage of FMM movements, backgrounds with the FMM closed are only taken executed in combination with other calibration measurements. These are scheduled daily/weekly/fortnight/monthly. As such not every cycle of 15 orbits contains an equal number of background measurements. During the E1 phase, scheduling also could be irregular. In addition, some issues with data transfer resulted in lost orbits. Hence the 40% requirement as background derivation with 1-2 orbits showed very poor results.*

SC23 So there are no shielded or non-illuminated pixels on the detector which can be

used for monitoring? *There are, but these are too few and showed too little variation within a single orbit to be usable.*

SC24 So there is a different ADC for the two sides of the detector? This should be mentioned. Are the ADCs located at different places? Is the effect included in the L1 offset correction?

*This is described in Hoogeveen et al., 2013. The effect is included (by default) in the offset correction.*

SC25 The definitions in line 9 and 12 of in-flight noise are not the same. *Corrected*

SC26 What are the criteria for 'proper 'and 'excessive'? The explanation is not any clearer than 'sufficient quality'. Please provide quantitative criteria. *For each quantity, the pixel is graded on a sliding scale. A combination of performances will determine if a pixel is flagged or not.*

SC27 Why are non-illuminated pixels described as non-functional? *This was an adopted definition. We expanded the text to be more clear*

SC28 Does this plot include the non-illuminated pixels? *No, we added a note in the text about this.*

SC29 So the non-illuminated pixels are included in these numbers? Please list them separately. *No, this was a mistake in the table . It was meant to exclude the non-illuminated pixels.*

SC30 The number of bad and dead pixels decreased compared to on-ground. Is this caused by detector annealing during the de-gassing? *That is possible. Although we cannot exclude other origins. These have been described now.*

SC31 *This has now been illustrated in section 2.*

SC32 On page 3 line 9 it is described that the DLED light does not pass through the grating, and now the grating is seen as the cause for the feature. How is this possible?

*The referee is correct that it is not possible. This has been changed.*

SC33 What do you refer to as "the complete light path within the module"? Where is the primary mirror? From the TROPOMI L1 ATBD I get the impression, that the primary telescope mirror is included in all optical paths for all sources (apart from the DLED) *The inclusion of a new figure in Section 2 should clear this up.*

SC34 From the page mps.tropomi.eu/calendar measurement search I find 4 WLS modes with three different exposure times for SWIR using DIFM SLS (4) or DIFM NOM (1). The longest exposure time I find is 88ms; this isn't entirely consistent with your statement. *That is correct. We rephrased this statement as the measurements with longer exposure times are not used in This paper. They are so long to accommodate usages of the WLS for UVN calibration.*

SC35 Please add that the stray-light is normalized to the peak value and that a spectral band as opposed to a spot (on-ground) is illuminated. *Straylight is normalized to total value, and not peak value. As that statement was also missing, it has been added.*

SC36 The measurements in Figure 5 are performed with oscillating diffuser? *Yes*

SC37 There are no examples provided on the in-depth analysis, so I would either remove this sentence or provide the examples. *Done*

SC38/SC39/SC40/SC43 *Yes, this was done on purpose. The dates in the text should have reflected that. It has been fixed.*

SC41 Large scale variations...: It might be useful to show how the module behaves in non-nominal situations. The data could be shown in a different colour in the same plot. Or is this inflating the scale? *This very strongly inflates the scale. It is also considered to be beyond the scope of this paper as the origins are known and characterized.*

SC42/SC45/SC49/SC50/SC51 *We disagree with these assessments. As the structure of the paper is set up in time, with results beyond orbit 2818 are included here and early results described earlier. The continuity in the figure illustrates the stability.*

SC44 Are all the numbers on dead and bad pixels without the shielded pixels? *Yes, the text has been changed to better describe this.*

SC46 Why should a relative irradiance factor change over time? Shouldn't that show in a seasonal effect? This hypothesis should be explained better or be removed. Did you exclude heating up of the calibration unit or offset effects? Usually degradation effects are observed to be proportional to exposure; however both diffusers seem to show the same slope. So a change in diffuser reflectivity with the same slope for both is unexpected. Do you observe any change with a similar slope for radiance?

*This section has been updated. However, seasonal effects of the irradiance should be accounted for, and not be included in the monitoring product. Note that an identical slope in both diffusers can be caused in degradation in the common path not related to the diffusers. It is too small to be quantified with the radiance.*

SC47/SC48 The DLED and the solar irradiance cover different optical paths, to exclude the detector responsivity cannot be concluded by the increase in solar irradiance. The increase could be much larger covering a decrease. Is there another means to exclude the detector? Maybe radiance data? Ok, the WLS confirms the DLED degradation. Does the WLS then exclude heating or degradation of the calibration unit?

*We are excluding the detector responsivity to be the main culprit for the DLED response. The increase seen in the solar irradiance can indeed be caused by improved detector responsivity, hiding behind the DLED degradation. The WLS could confirm this, if it was less variable. It, in combination with the temperature info from the Monitoring results, does appear to exclude heating of the calibration unit. Radiance data may be usable, but was considered to be beyond the scope of this paper. This will be done in a future publication.*

SC52 Can the outlier around orbit 7000 be explained? No, it cannot.

SC53 *Only a few events took place. Given the timescale and scheduling of calibration*

*measurements around these, no CKD derivation can be done. As such, we can only take a look at recovery in days after this event. This was included and no outliers have been found (note that orbit 7000 is not this event and therefor cannot be explained.)*

SC54 *Changes to the processor will be described in a future paper by Ludewig et al., The word 'ample' is a language issue. This has been rephrased.*

Formatting: *(Note that all new figures are indexed one higher due to the insertion of a new figure 1.)* Fig 2. And 3. *This was tried, but we did not get a satisfying figure.* Fig. 4,5, 7, 9, 10 and 12: *We prefer to use value and explain the exact meanings in the caption. A change in these The term very quickly causes confusion.* Fig. 6 *changed*

Fig. 16/17: *This is correct, but we are not plotting the same thing with in-flight being red and reference being blue (see Fig. 15). For Figure 16 a choice had to be made to either reference Fig. 17 in-flight or Fig. 15 in-flight. We choose Fig. 15.*

Typos *All were corrected if they were still present in the text.*

---

## Author Comment (AC3) · 1 Nov 2019

*We would like to thank the referee for their time and effort in reading and reviewing our paper with constructive comments. Please find our answers to your comments below in italics*

Figure 4: There's no mention of the linear features in the radiances at indices 400 and 500 - are these dark flux or something else? Is there a mask on the detector for the top and bottom rows? The results look very different compared to the middle of the detector.

*The line at index 400 is an unexplained higher dark flux (now referred to as dark current, see response to the other referee). Note the higher dark current is only 5-10%. The*

[Figure]

*uncertainty in the dark current fit is much higher. The line at index 500 is the edge of the ADC, also causing a higher dark current. Note that both increases are significantly lower than the useful signal.*
*The top and bottow rows are indeed covered and not illuminated*

Page 8, Line 3-4: Did they not have any thermal measurements for an actual comparison? This difference may be important.

*The measurements during the on-ground calibration were taken at a slightly different instrument temperatures. This explains the small differences observed. measurements with different thermal conditions were scheduled for the in-flight E1 phase, but not executed.*

Figure 7: There's a pretty distinct discontinuity in the offset at the middle of the detector, but I don't see any discussion of this in the text.

*This is mentioned on page 12, lines 3-5. Note that the discontinuity is at the level of 0.3 mV, so less than 0.1% of the absolute offset.*

Figure 8: There seems to be a linear trend in the median offset, but that could be an artifact of the statistics rather than a real thing. Which is it?

*This is indeed an artifact of the statistics. The longer time period -as shown in figure 19- does not show a linear trend.*

Figure 11: How is it possible that the number of bad/dead pixels decreases with time? Did your criteria change?

*There are a few reasons. First, the thermal stability of the various components during on-ground calibration was much worse then it is now in-flight. This causes a much better characterization of the noise behavior of each pixel in-flight due to the better thermal stability in-flight. However the measurement uncertainty of the noise is worse due to fewer measurements. Second, noise/RTS behavior of individual pixel change after annealing of the detector. Annealing automatically occurs when the detector is*

*heated up to room temperature and subsequently cooled down again. Annealing occured between OCAL and launch and several times during E1 due to the specific commissioning planning. An additional annealing took place during orbit 3500 due to a spacecraft anomaly. The decrease seen in Fig. 11 is most likely caused by the start of nominal operations, which stabilized the thermal environment as no more 'exotic' calibration measurements were taken.*

Figure 17 shows that this could be a 1% error, which is significant for trace gas retrievals.

*There is a difference between a 1% error in the actual ISRF and a 1% change in the SLS response using a stationary or oscillating diffuser. A stationary diffuser introduces speckle patterns. It is these speckle patterns that introduce the difference shown here. This is not in the ISRF itself. Uncertainty in the ISRF is much lower and discussed in van Hees et al., 2018.*

Page 17, Line 5-6: This is really only a measurement of a small area of the bandpass. Is there any reason to assume that this is representative of all of the other wavelengths as well? Please note that I realize this is a heroic effort to track stray light.

*We can cover five small spectral regions of the bandpass, which are regularly spaced. These all show very similar patterns and all confirm the assumed uniform straylight behavior previously measured on-ground with external light sources (see Tol et al., 2018). As far as we can tell from these bandpasses it is representative for the entire detector. It is possible the straylight is not representative anymore for a small bandpass in between the five probed bandpasses, without affecting the measured bandpasses. However, we consider this effect to be highly unlikely. Given typical origins of changes in straylight most likely are large-scale (with the exception of ghosts).*

What is the difference between Figures 16 and 17?

*Figure 16 shows the range of measurements probing all the rows. This shows the*

*median and the 1 and 99% of each row. Figure 17 shows the two medians (which are indeed the same as the medians in Figure 16) and the residual. This was first presented in a single figure, but the figure became too information-dense.*

Page 26, Line 2: I disagree, the offset image in Figure 7 shows a pretty clear difference.

*Given the scales I respectfully disagree. We did rephrase it by adding the word 'significantly'.*
* * *

---

## Author Comment (AC4) · 1 Nov 2019

*We would like to thank the referee for their time and effort in reading and reviewing our paper with constructive comments. Please find our answers to your comments below in italics*

General Comments: As most of the issues in this paper were already covered by the comments of anonymous referees 1,2 and 3, I will mostly focus on issues not sufficiently covered by the other reviewers. I agree with referee 3, that the techniques discussed in this manuscript have already been presented elsewhere in the literature and therefore are not novel, but the docu- mentation of the TROPOMI instrument status is in itself a useful and important work. Therefore I also recommend publication after

[Figure]

minor corrections.

As however the manuscript seems largely to be extracted from a status report, I would suggest changing the title of the manuscript to "Technical Note: In-flight calibration and monitoring of the TROPOMI-SWIR module".

*We have contacted the editor about this suggestion.*

I also recommend adding consistent captions to all figures (e.g. figure 6: [e-/sec] is missing, figure 8: electronic offset is reported in V, rather than in [e-] and therefore magnitude and variation of the offset could not be compared quickly to the magnitude of the DC variation).

*This was done, see also other Author comments*

I also wonder, why the eclipse side of the Earth was assumed pre-launch as dark, as black body radiance at âĹij 293K is non zero in the spectral range > âĹij 1750 nm. There- fore it is not astonishing, that the TROPOMI-SWIR module detects thermal radiation of the earth with the FFM open.

*Although it is indeed not surprising, the variation of the thermal radiation (i.e. the ease with which differences between 280 and 293 K could be detected, as well as the presence of point sources)*

When discussing the DC, it should be mentioned that the DC consists of two primary components, Detector DC and DC introduced by the thermal radiation of the instruments optical bench itself.

*This is included in a footnote now.*

Reviewer 3 also asks if there is a mask implemented on the detector. Some literature research (i.e. Paul et al., Characterization and correction of stray light in TROPOMI-SWIR) reveals, that "In the grey area, the light is blocked by the entrance slit of the spectrometer (top and bottom) or a shield at the detector (left and right)", see caption

"Figure 2" of this manuscript. As shielded rows represent an excellent possibility to disentangle detector dark current from ambient dark current (produced from the optical bench), I would strictly recommend expanding the DC analysis including that dataset, given that such data is available. Such data could also reveal potential problems with dark signal shifts known to exist when illuminating larger parts of MCT-SWIR detectors, see for instance: Chapman et. al, "Spectral and Radiometric Calibration of the Next Generation Airborne Visible Infrared Spectrometer (AVIRIS-NG)", and according explanation of pedestrial shift.

*This was considered. However, the shields and areas around the entrance slit are kept at the similar temperatures as the optical bench and the detector DC was found to vary over larger or at least similar scales as any temperature difference (see Fig. 5, prey. Fig. 4). Hence disentangling the contributions from the detector dark current and ambient dark current has proven to be non-trivial.*

As mentioned by Paul et al. light on the SWIR detector is also blocked by the entrance slit on top and bottom of the spectrometer. Typically, such measurement could be used to assess the In-Band stray light level of the instrument. Therefore I recommend to add an according analysis, if such data is available for the TROPOMI SWIR spectrometer module.

*This was considered, but found to be beyond the capability of the available data. It was found that such an analysis would only work for straylight in the spectral directions and with the availability of only a diode laser (SLS) instead of a point source (as used in Tol et al.) this analysis would be unable to distinguish from purely spectral and spectral-spatial straylight. Also, the location of the nearest diode lasers to the edges of the spectral slit are still significant.*

Can you furthermore give more explanation on the possible cause of the fringes observed in Fig. 12, top (DLED measurements) ?

*This was added.*

In addition I would recommend adding sketches of the optical path for the different measurement configuration as these will ease up to follow the different optical configurations mentioned in the manuscript.

*A new figure 1 with accompanying text was added.*